# Evaluating multiple models using labeled and unlabeled data

**Divya Shanmugam[1]*   Shuvom Sadhuka[2]***
**Manish Raghavan[2,3]   John Guttag[2]   Bonnie Berger[2]**   Emma Pierson[4]****

[1]Cornell University
[2]Massachusetts Institute of Technology
[4]University of California, Berkeley
*Equal contribution, **Equal supervision

divyas@cornell.edu   ssadhuka@mit.edu

## Abstract

It is difficult to evaluate machine learning classifiers without large labeled datasets, which are often unavailable. In contrast, *unlabeled* data is plentiful, but not easily used for evaluation. Here, we introduce Semi-Supervised Model Evaluation (SSME), a method that uses both labeled *and* unlabeled data to evaluate machine learning classifiers. The key idea is to estimate the joint distribution of ground truth labels and classifier scores using a semi-supervised mixture model. The semi-supervised mixture model allows SSME to learn from three sources of information: unlabeled data, multiple classifiers, and probabilistic classifier scores. Once fit, the mixture model enables estimation of any metric that is a function of classifier scores and ground truth labels (e.g., accuracy or AUC). We derive theoretical bounds on the error of these estimates, showing that estimation error decreases with the number of classifiers and the amount of unlabeled data. We present experiments in four domains where obtaining large labeled datasets is often impractical: healthcare, content moderation, molecular property prediction, and text classification. Our results demonstrate that SSME estimates performance more accurately than do competing methods, reducing error by $5.1\times$ relative to using labeled data alone and $2.4\times$ relative to the next best method.

## 1  Introduction

Large, labeled datasets are critical for evaluation in machine learning, but such datasets are often prohibitively expensive or impossible to obtain [17, 24]. Exacerbating the challenge of evaluation, the number of off-the-shelf classifiers has increased dramatically through the widespread usage of model hubs. The modern machine learning practitioner thus has a myriad of trained models, but little labeled data with which to evaluate them.

In many domains, *unlabeled data* is much more abundant than labeled data [8, 61, 49]. To take advantage of this, we introduce Semi-Supervised Model Evaluation (SSME), a method that can be used to evaluate multiple classifiers using both labeled *and* unlabeled data. The key idea is to estimate the joint distribution of ground truth classes $y$ and continuous classifier scores $\mathbf{s}$ using a mixture model, where different components of the mixture model correspond to different classes. The joint distribution allows us to evaluate performance on examples where we have access *only* to each classifier's scores, i.e. unlabeled examples. SSME can estimate any metric that is a function of class labels and probabilistic predictions, including widely-used metrics like accuracy, expected calibration error, AUC, and AUPRC. Concretely, we can evaluate a classifier on an unlabeled point by using our estimate of $P(y, \mathbf{s})$.

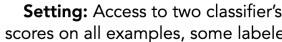

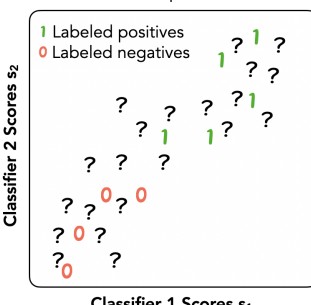
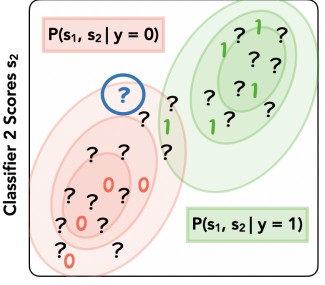
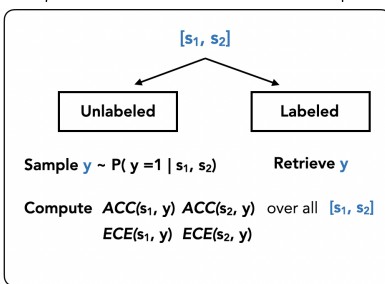

Figure 1: **Using SSME with two binary classifiers.** SSME applies to settings in which classifier scores are available for a set of examples, some labeled (left). SSME first fits a mixture model to estimate the joint density of scores and labels (where $s_1$ is the score assigned by the first classifier, and $s_2$ the second, middle panel). SSME next uses the mixture model to compute metrics with respect to the expected label for each example (right). SSME can be applied to any number of classifiers and classes $K$, and supports any metric that compares classifier scores to ground truth labels.

SSME is the first evaluation method to capture three key facets of modern machine learning settings: (i) multiple machine learning classifiers, (ii) probabilistic predictions over all classes, and (iii) unlabeled data. Simultaneously using all three is difficult because it requires accurately estimating the (potentially high-dimensional) joint distribution $P(y, \mathbf{s})$ with primarily unlabeled data. While prior work captures subsets of these three sources of information [69, 57, 37, 15, 11] — for example, augmenting labeled data with unlabeled data to evaluate a *single* classifier — no existing approach accommodates all three.

We show both theoretically and empirically that by using all available data — multiple classifiers, continuous scores over all classes, and unlabeled data — SSME is able to produce more accurate performance estimates compared to prior work. We make three contributions:

1. We propose SSME, a method that incorporates unlabeled data, continuous classifier scores, and multiple classifiers to produce more accurate performance estimates than prior work. SSME is able to estimate any metric that compares predicted probabilities to ground truth labels and accommodates varying numbers of classifiers and classes.
2. We prove that using unlabeled data and multiple classifiers improves performance estimation, providing insight into SSME's strong empirical performance across domains.
3. We show that SSME produces more accurate performance estimates compared to labeled data alone and seven baselines, reducing metric estimation error by $5.1\times$ relative to labeled data and $2.4\times$ relative to the next best method. We further validate SSME's utility in two case studies: evaluating large language models and subgroup-specific performance.

## 2  Problem Setting

We consider a setting in which a practitioner wishes to evaluate several classifiers using both labeled and unlabeled data. Formally, there are $M$ classifiers $[f_1, f_2, \ldots, f_M]$ designed for the same task, which might differ in their training data, architectures, or optimization, among other possibilities. Each classifier maps from the same input domain $\mathcal{X}$ to a probability distribution over $K$ classes. Let $\mathbf{s}^i = [f_1(x^i), f_2(x^i), \ldots, f_M(x^i)]$ denote the concatenated set of classifier scores for data point $x^i$; Figure 1 depicts the setting.

We assume access to the set of classifiers and two datasets: (1) a small labeled dataset, $\mathcal{D}_L = \{(x^i, y^i)\}_{i=1}^{n_\ell}$ and (2) a larger unlabeled dataset $\mathcal{D}_U = \{(x^i)\}_{i=1}^{n_u}$. We assume that unlabeled data is far more available than labeled data, i.e. $n_u >> n_\ell$, and that the unlabeled samples are drawn from the same distribution as the labeled samples. Our goal is to use these two datasets to estimate classifier performance using standard metrics such as expected calibration error (ECE) or accuracy. If we knew the true label $y^i$ for each point $x^i$, it would be straightforward to evaluate the performance of each pre-trained classifier. However, the true label is not available for unlabeled examples, so we instead aim to infer a distribution over ground truth labels.

# 3  Method

Semi-supervised model evaluation (SSME) contains two key steps. First, SSME estimates the joint distribution of ground truth labels $y$ and classifier scores $\mathbf{s}$ — i.e. $P(y, \mathbf{s})$ – using both labeled and unlabeled data. Second, SSME uses the estimate of $P(y, \mathbf{s})$ to compute metrics with respect to the posterior over $y$ implied by classifier scores $\mathbf{s}$, allowing classifier evaluation using both labeled data (where ground truth is available) and unlabeled data (where ground truth is inferred).

**Step 1: Estimating $P(y, \mathbf{s})$.**  SSME estimates $P(y, \mathbf{s})$ using a semi-supervised mixture model, where components of the mixture model correspond to different ground truth classes. Following standard procedures in mixture model estimation, we optimize parameters of the mixture model $\theta$ by maximizing the log-likelihood of the observed data. When $y$ is unobserved, we treat it as a latent variable and marginalize it out. Specifically, we optimize the following expression:

$$\operatorname*{argmax}_{\theta} P(\mathbf{S}, Y; \theta) = \operatorname*{argmax}_{\theta} \left[ \prod_{i=1}^{n_\ell} P(\mathbf{s}_i, y_i; \theta) \prod_{j=1}^{n_u} P(\mathbf{s}_j; \theta)^{\lambda_U} \right] \tag{1}$$

$$= \operatorname*{argmax}_{\theta} \left[ \underbrace{\sum_{i=1}^{n_\ell} \log \left[ P_\theta(\mathbf{s}_i | y_i) P_\theta(y_i) \right]}_{\text{Labeled Data Log-Likelihood}} + \lambda_U \underbrace{\sum_{j=1}^{n_u} \log \sum_{k=1}^{K} \left[ P_\theta(\mathbf{s}_j | y_j = k) P_\theta(y_j = k) \right]}_{\text{Unlabeled Data Log-Likelihood}} \right] \tag{2}$$

where $\lambda_U$ controls the relative weight of the unlabeled data in the likelihood; we fix $\lambda_U = 1$.

We optimize the above expression using expectation-maximization (EM) [19, 76], alternating between the E-step, which estimates which mixture component (i.e. ground truth class) each data point belongs to, and the M-step, which estimates the parameters of each component based on the soft component assignments. Algorithm 1 details the EM procedure we use to fit our mixture model.

A technical challenge when fitting densities on bounded domains, as is the case with classifier outputs, is *boundary bias*, where estimates of density are biased near edges of the domain [39]. To overcome this challenge, we transform probabilistic predictions over $K$ classes to points ("scores") in $\mathbb{R}^{K-1}$ using the additive log-ratio transform, which produces a one-to-one mapping between the two spaces [1, 53]. Additive log-ratio transforms are a *reparameterization trick*: they transform the classifier scores from a hard-to-model space — the probability simplex — to an easier-to-model space — unbounded reals. Appendix E.1 provides the expression for the additive log-ratio transform.

**Step 2: Estimating classifier performance given $P(y, \mathbf{s})$.**  Once we have estimated the parameters $\theta$ for the mixture model, we can use the fitted density to estimate metrics of interest, using the procedure described in Figure 1. By estimating the conditional distribution $P_\theta(y | \mathbf{s})$, SSME can measure performance with respect to the *expected* label of unlabeled examples (where we observe only $\mathbf{s}^i$). Specifically, we sample ground truth labels $y$ from the distribution $P_\theta(y | \mathbf{s})$ for all unlabeled examples. Labeled examples retain their ground truth labels. We then compute metrics with respect to both the sampled and ground truth labels, and average the estimated metric across samples. SSME can estimate any metric that is a function of classifier scores and ground truth labels; see Algorithm 2 for corresponding pseudocode.

The proposed approach enjoys several properties. By using a semi-supervised mixture model, SSME takes advantage of unlabeled data to estimate $P(y, \mathbf{s})$. SSME also naturally accommodates and benefits from additional classifiers, because they can provide additional information about the ground truth label of an unlabeled example (additional classifiers simply translate to additional dimensions in $\mathbf{s}$). Furthermore, the use of the additive log-ratio transform allows SSME to learn directly from continuous class probabilities for each input, as opposed to discretized labels.

## 3.1  Implementation

SSME can accommodate multiple ways to parametrize the class-conditional distribution of scores $P(\mathbf{s} \mid y)$ as long as they can be learned in the semi-supervised setting described above. We denote the parameterized distribution as $P_\theta(\mathbf{s} \mid y)$. In our experiments, we use a kernel density estimator (KDE) to parameterize $P_\theta(\mathbf{s} \mid y)$ and explore alternative parametrizations of $P(\mathbf{s}^{(i)} | y^{(i)})$ in Appendix E.3.

**Algorithm 1:** Estimating $P(y, \mathbf{s})$ using EM

---

**Input:** Labeled set $\mathcal{D}_\ell = \{(\mathbf{s}^i, y^i)\}_{i=1}^{n_\ell}$; Unlabeled set $\mathcal{D}_u = \{\mathbf{s}^i\}_{i=1}^{n_u}$; $\mathbf{s}^i$ are ALR-transformed classifier scores; Labeled-data weight $\lambda_U$; number of classes $K$; total epochs $T = 1000$

**Init:** Initialize model parameters $\theta^0$;

Initialize class priors $p^0(y = k) = \hat{p}(y = k)$ ;                     // empirical class distribution

1   **for** $t \leftarrow 0$ **to** $T - 1$ **do**

2     **E-step: compute responsibilities** $\gamma_{ik}$;

3     **foreach** $\mathbf{s}^{(i)} \in \mathcal{D}_u$ **do**

4       **for** $k \leftarrow 1$ **to** $K$ **do**

5         $\gamma_{ik} \leftarrow \dfrac{p^t(y = k)\, P_{\theta^t}(\mathbf{s}^{(i)} \mid y = k)}{\sum_{\ell=1}^{K} p^t(y = \ell)\, P_{\theta^t}(\mathbf{s}^i \mid y = \ell)}$;

6     **foreach** $(\mathbf{s}^i, y^i) \in \mathcal{D}_\ell$ **do**

7       $\gamma_{ik} \leftarrow \mathbf{1}[k = y^i]$ ;                     // fix to true label

8     **M-step: update priors and** $\theta$;

9     **for** $k \leftarrow 1$ **to** $K$ **do**

10      $N_k \leftarrow \sum_{\mathcal{D}_u} \gamma_{ik} + \lambda_U \sum_{\mathcal{D}_\ell} \gamma_{ik}$ ;                  // weighted effective count for class $k$

11      $p^{t+1}(y = k) \leftarrow N_k / (n_u + \lambda_U n_\ell)$;

12     $\theta^{t+1} \leftarrow \arg\max_\theta \sum_{i=1}^{n_u + n_\ell} w_i \sum_{k=1}^{K} \gamma_{ik} \log P_\theta(\mathbf{s}^i \mid y = k),$

13     where $w_i = \lambda_U$ for unlabeled points and $w_i = 1$ otherwise;

**Output:** Final parameters $\theta^T$ and priors $p^T(y)$

---

Kernel density estimators do not make parametric assumptions about the distributional form of each component: this is useful for modeling distributions of predictions, which can vary widely across tasks and classifiers. The KDE for the $k$th class is parametrized by bandwidth $h$ and the kernel type $\mathcal{K}$. We weight each point by the probability it belongs to the component (i.e., a soft label). When $\mathbf{s}^i$ is labeled, $P(y^i = k) = 1$ for the true label $k$, and when $\mathbf{s}^i$ is unlabeled, we compute $P(y^i = k|\mathbf{s})$ using the estimated density. For all experiments, we fit use a Gaussian kernel and estimate $h$ using the improved Sheather-Jones algorithm [10]. We optimize the parameters using EM over 1000 epochs. We initialize component assignments by drawing a label for a given example according to the mean classifier score across the set of classifiers.

### 3.2 Theoretical analysis

We motivate our approach by deriving bounds for how well a semi-supervised estimation procedure can estimate classifier performance in a setting where classifier scores $\mathbf{s}$ are drawn from a mixture of Gaussians.

**Model of $y$ and $\mathbf{s}$.** We consider a stylized binary classification setting drawn from previous work [67] where classifier scores $\mathbf{s}$ are drawn from a balanced mixture of two Gaussians. For data in the positive class, classifier scores are drawn from a multivariate Gaussian $\mathcal{N}(c, I)$. For data in the negative class, classifier scores are drawn from a multivariate Gaussian $\mathcal{N}(0, I)$. Let $n_u$ refer to the number of unlabeled examples, $n_\ell$ the number of labeled examples, and $d$ the number of classifiers. Additional classifiers increase the dimensionality of the multivariate Gaussian.

**Performance estimation procedure.** Having specified how classifier scores are generated, we analyze how accurately a semi-supervised performance estimation procedure similar to our own can estimate classifier performance. We consider a class of semi-supervised learning algorithms frequently analyzed in prior work, UL+ [67]. UL+ algorithms first use the unlabeled data to identify decision boundaries and then use the labeled data to assign regions to classes. $AUC_k$ and $ACC_k$ represent ground truth performance of classifier $k$; $\widehat{AUC}_k$ and $\widehat{ACC}_k$ are performance estimators that use UL+ to estimate a Gaussian mixture model, and use that model to estimate performance. We apply bounds developed by [67] for errors in mixture model estimation to bound errors in *performance* estimation.

**Theorem 1** Let $\widehat{AUC}_k$, $\widehat{ACC}_k$ be estimated using a semi-supervised learning algorithm in the UL+ class of estimators. When $n_u \gtrsim \max\left\{d, \frac{d}{||c||^4}, \frac{\log(n_u)}{\min\{||c||^2, ||c||^4\}}, \frac{d\log(dn_u)}{||c||^6}\right\}$ and $d \geq 2$, the following bounds on error in estimated performance hold with probability $1 - p$:

$$|AUC_k - \widehat{AUC}_k| \leq \quad \Phi\big(\frac{\mathbf{c}_k}{\sqrt{2}}\big) - \Phi\big(\frac{\mathbf{c}_k - \epsilon_\mathbf{c}}{\sqrt{2}}\big) \tag{3}$$

$$|ACC_k - \widehat{ACC}_k| \leq \quad \Phi\big(\frac{\mathbf{c}_k}{2}\big) - \Phi\big(\frac{\mathbf{c}_k - \epsilon_\mathbf{c}}{2}\big) \tag{4}$$

where $\epsilon_\mathbf{c} \lesssim \frac{1}{p}\left(\sqrt{\frac{d}{||c||^2 n_u}} + ||c||e^{-\frac{1}{2}n_l||c||^2\left(1 - \frac{C_0}{||c||^2}\sqrt{\frac{d\log(n_u)}{||c||^2 n_u}}\right)^2}\right)$ and $C_0$ is a universal constant.

The expression for $\epsilon_\mathbf{c}$ captures its asymptotic dependence on $n_u$, $n_\ell$, $d$, and $c$. Smaller $\epsilon_\mathbf{c}$ correspond to smaller errors in performance estimation. Section C contains the full proof, which consists of two steps: we first bound the error in estimating the separation of the two Gaussian components $||\mathbf{c}||$, and then use the relationship between component separation and classifier performance to bound error in estimates of AUC and accuracy.

**Analysis.** Our theoretical bounds have several implications. First, unlabeled data helps: increases in $n_u$ reduce both the first term and the second term. This indicates why SSME outperforms baselines that only make use of labeled data: additional unlabeled data improves our ability to accurately estimate components of a mixture model ($\epsilon_\mathbf{c} \approx O(1/\sqrt{n_u})$). Second, bounds tighten as separation between components $||\mathbf{c}||$ grows, i.e. when classifiers in the set are more accurate. Third, bounds tighten as the number of classifiers increases, so long as the increase in separation $||c||$ outweighs the cost of an additional dimension in $d$; we precisely characterize this setting in Section C.1. This helps explain why SSME outperforms work that makes use of a single classifier, instead of several.

Our theoretical results are derived for a stylized setting frequently studied in prior work [59, 60, 67, 26]. While this setting differs from our empirical setting – for example, it uses a simplified semi-supervised learning algorithm – we verify that trends our theory implies hold robustly in both synthetic and real experiments (Sec. 5, D.1), including settings where these assumptions do not hold.

## 4 Experiments

### 4.1 Datasets and Classifier Sets

We select datasets and classifier sets to be realistic and diverse, capturing multiple modalities (EHRs, text, and graphs), domains (healthcare, content moderation, chemistry), and architectures (logistic regressions, graph neural networks, large language models). We report ground truth metrics for the binary and multiclass classifiers in Tables S1 and S2 respectively. We also include a detailed description of each dataset and classifier set in Appendix B.1. We summarize each dataset and differences between classifiers in the associated classifier set below.

We evaluate SSME on five classification datasets: (1) **MIMIC-IV** [38], a dataset of Boston-area electronic health records (EHRs) with three hospitalization outcomes (critical outcome, ED revisit within 30 days, and hospital admission) with classifiers drawn from prior works [50]; (2) **Civil-Comments** [9], a dataset of social media comments flagged as "toxic" or not by human annotators, with pretrained classifiers downloaded from the WILDS benchmark [42]; (3) **OGB-SARS-CoV** [33], a molecular property prediction task, with pretrained classifiers downloaded from WILDS; (4) **MultiNLI**, a natural language task, with classifiers drawn from the SubpopBench dataset [73]; and (5) **AG News** [75], a news classification task, using open-source and closed-source LLMs as classifiers.

### 4.2 Baselines

We compare SSME to eight baselines that use both labeled and unlabeled data to estimate performance (with the exception of **Labeled**, the standard approach to classifier evaluation, which only uses examples for which labels are available). The remaining baselines cover a broad range of approaches one could adapt to semi-supervised evaluation and fall into three categories. For a full description of each baseline's implementation, please refer to Appendix B.2.

The first category fits a model of $P(y, \mathbf{s})$ using labeled data alone, then applies it to the entire dataset . Each baseline does so in a different way: Pseudo-Labeled (**PL**) fits a logistic regression to predict the true label from classifier scores; Bayesian-Calibration (**BC**) [37] fits a Bayesian model to re-calibrate each classifier's predicted probabilities; **AutoEval** [11] learns a function to debias classifier predictions. The second category estimates the performance of individual classifiers. **SPE** [69] fits a parametric mixture model to a single classifier's predictions. **Active-Testing** [44] uses active learning to select examples to label out of a pool of unlabeled examples based on a single classifier's predictions and evaluates the classifier on those examples. The last category of baselines is drawn from the multiple annotator literature, where annotations are assumed to be discrete. We compare to **Dawid-Skene** [18], which accepts discrete annotations from each classifier to estimate the latent true label for each example. We also compare to **Majority-Vote**, which ensembles classifier predictions by performing an accuracy-weighted aggregation of classifier predictions.

We also provide a comparison to completely unsupervised approaches (including ensembling where the "ground truth" label is drawn based on the average classifier score) in Sec. D.10, and discuss connections to weak supervision in Sec. D.9.

### 4.3 Evaluation

We evaluate SSME's ability to estimate four performance metrics for binary classifiers: accuracy, area under the receiver operating characteristic curve (AUC), area under the precision-recall curve (AUPRC), and the expected calibration error (ECE). For multi-class problems, we evaluate accuracy and top-label calibration error [32].

We partition each dataset into three splits: the classifier training split (used to train the classifiers whose performance SSME estimates), the estimation split (used to fit SSME and estimate classifier performance), and the evaluation split (used to produce a held-out, ground-truth measure of classifier performance). All splits are sampled from the same distribution, except when estimating subgroup-specific performance, where the evaluation split contains a subset of the test distribution.

To evaluate metric estimates, we measure the absolute error of the *estimated* metric, computed using the estimation split, compared to the *true* metric, computed on the held-out evaluation split (averaging over classifiers in the set). The estimation split consists of either 20, 50, or 100 labeled examples and 1000 unlabeled examples across all experiments. The size of the evaluation split is on the order of thousands of labeled examples and varies by task (see Appendix B.1 for exact split sizes). To quantify uncertainty, we compute 95% confidence intervals using the standard error of the mean across 50 random data splits ($\pm 1.96 \times$ SEM). In line with prior work, we report rescaled estimation error for each metric (where errors are relative to using labeled data alone), allowing us to standardize the scale of errors across datasets and metrics [28].

## 5 Results

### 5.1 SSME produces more accurate performance estimates than prior work

We now compare SSME to eight baselines in terms of its ability to estimate classifier performance on five binary tasks. All figures report rescaled metric estimation error (RMAE; lower is better) and reflect performance estimation using 20 labeled examples and 1000 unlabeled examples. Results are consistent across additional values of $n_\ell$ (50 and 100; see results in D.2), although labeled data grows more competitive (as expected) with larger labeled dataset sizes.

**Comparison to baselines**  SSME achieves lower mean estimation error (averaging across tasks and metrics) than all baselines, indicating more accurate estimation of classifier performance. SSME reduces estimation error by $5.1\times$ relative to labeled data alone (averaged across tasks and metrics). In contrast, the next best method reduces estimation error by $2.4\times$. SSME also outperforms baselines on specific metrics. For accuracy, SSME reduces metric estimation error, relative to using labeled data alone, by $5.6\times$ (averaged across tasks); the next best method for each dataset reduces metric estimation error by $2.0\times$. While the magnitude by which SSME beats baselines varies —for example, SSME reduces error by $2.9\times$ on AUC (averaged across tasks), while the next best method reduces error by $2.6\times$ — SSME consistently outperforms baselines across metrics. SSME performs on par with or better than the strongest baseline in 51 of 60 dataset, metric, and labeled data combinations

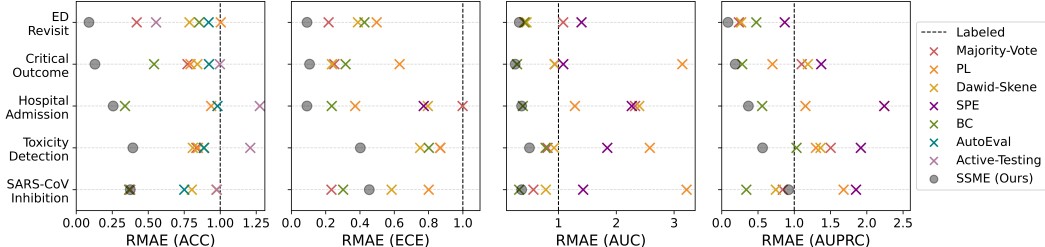

Figure 2: **Metric estimation error on binary tasks ($n_\ell = 20, n_u = 1000$).** Each point plots the rescaled mean absolute error (RMAE) across runs, where 1.0 (dashed line) represents the RMAE when using labeled data alone and lower is better. SSME (gray) achieves lower estimation error than the baselines, averaging across metrics and 50 runs, reported for five binary tasks (y-axis). Results reporting mean absolute error (unscaled) and 95% CIs are in Tables S3-S6.

we study, and significantly better in 31 settings (Tables S3-S6). Figure 2 summarizes how SSME compares to several baselines across various metrics and tasks[1].

Our results are also encouraging in absolute terms. With 20 labeled examples and 1000 unlabeled examples, SSME estimates accuracy within 1.5 percentage points (averaging across tasks). The closest baseline estimates accuracy within 3.4 percentage points. Results comparing SSME to the next best baseline on other metrics (1.9 vs 3.8 on ECE; 3.6 vs 4.3 on AUC; 8.5 vs 10.2 on AUPRC) confirm that SSME not only achieves more accurate classifier performance estimation compared to prior work, but that the resulting performance estimates are reasonably close to the ground truth.

Consistent with our theoretical results, SSME estimates performance more accurately because it makes use of all available information: multiple classifiers, continuous scores over all classes, and unlabeled data. *Dawid-Skene* and *AutoEval* discretize classifier scores (although *AutoEval* does make use of classifier confidence associated with each discrete prediction). *Pseudo-Labeling*, *Bayesian-Calibration*, and *AutoEval* each learn a mapping from $\mathbf{s}$ to $y$ using only the labeled data and apply that mapping to the unlabeled data (rather than learning from labeled and unlabeled data together). By jointly learning across both labeled and unlabeled data, SSME is able to generalize much better in cases where there aren't enough labels to estimate the joint distribution of classifier scores and labels from labeled examples alone. Finally, *Bayesian-Calibration* and *SPE* learn from a single classifier's scores, and do not learn from multiple classifiers at once.

**Comparison across metrics**    SSME provides the greatest benefits relative to labeled data alone when measuring expected calibration error (ECE), with a reduction in estimation error of $7.2\times$ (averaging across tasks). ECE is harder to estimate with few labeled examples because it requires binning and then averaging calibration error across bins, producing greater variability when the number of labeled points per bin is small. We observe the smallest benefits relative to labeled data alone when measuring AUPRC (a reduction in estimation error of $2.2\times$, relative to labeled data).

**Comparison across amounts of labeled data**    SSME's performance continues to improve with more labeled data, but the advantage it confers over labeled data decreases: for example, with 20, 50, and 100 datapoints, SSME outperforms labeled data alone by $5.6\times$, $3.0\times$, and $1.6\times$. Similar to labeled data alone, there are diminishing but positive returns to adding labeled data to SSME's performance estimation procedure.

Another way to quantify SSME's benefit is to measure the amount of labeled data required to match SSME's performance, or the *effective sample size* (ESS), as introduced by prior work [11] (see Appendix B.3 for implementation details). With access to 20 labeled examples and 1000 unlabeled examples, SSME achieves an average ESS of 539 labeled examples for estimating ECE (averaging over tasks). In contrast, the next best approach achieves an ESS of 110 labeled examples.

**Comparison to marginal fit**    To validate the benefit of fitting the mixture model to multiple classifiers simultaneously, we compare to an ablated version of SSME fit on a single classifier at a time, SSME-M. SSME-M estimates the classifier-specific marginal distribution of $P(y|\mathbf{s})$, and uses

---

[1]Fig. 2 omits results for Dawid-Skene (accuracy) and Majority-Vote (accuracy) on hospital admission, and SPE (ECE) because their higher RMAEs distort the plotting scale. Tables S3 - S6 contain complete results across baselines, including uncertainty estimates.

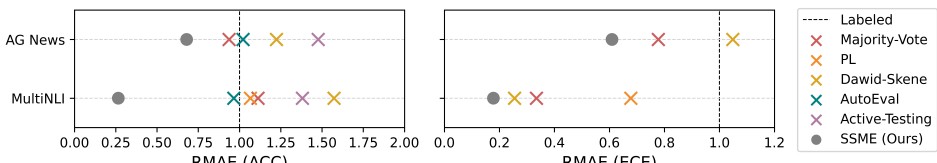

Figure 3: **Metric estimation error for LLM classifiers ($n_\ell = 20$, $n_u = 1000$).** SSME consistently reduces estimation error relative to labeled data alone and does so more reliably than any baseline capable of estimating performance in multiclass settings.

this estimate to evaluate the classifier in question. Doing so results in worse performance estimates across metrics, tasks, and amounts of labeled data (Tables S3- S6) relative to our full model. This agrees with our theoretical findings: each classifier provides distinct information about the ground truth label for a given example, which SSME is able to use.

**Supplementary results**   The supplement contains additional results. We report mean absolute error, including standard errors and additional $n_\ell$, for accuracy, ECE, AUC, and AUPRC in Tables S3- S6 (broken down by classifier in Tables S7- S10). We report the sensitivity of SSME to kernel choice in Table S13. Section D.11 provides a computational cost analysis of SSME relative to the baselines. Finally, we provide synthetic experiments to substantiate our theoretical findings under violations of our assumptions in Section D.1.

## 5.2   Case study: Evaluating classifiers derived from large language models

Large language models (LLMs) are increasingly used to create classifiers [40, 66, 68], with demonstrated potential to serve as data annotators [78]. However, evaluation of LLM-based classifiers is limited by the availability of labeled data. We study SSME's ability to estimate performance in this setting.

We evaluate classifiers derived from large language models on the MultiNLI and AG News datasets: the MultiNLI classifier set contains off-the-shelf language model based classifiers, while the AG News classifier set contains four classifiers trained atop open- and closed-source large language models. Fig. 3 reports our results. We find that SSME improves over labeled data by 2.3× (averaging over tasks and metrics). In contrast, the next best baseline (Majority-Vote) improves over labeled data by

|  | Age | Sex | Race |
|---|---|---|---|
| PL | 0.92 | 1.06 | 1.13 |
| Dawid-Skene | 0.75 | 0.67 | 0.67 |
| BC | 0.45 | 0.33 | 0.41 |
| SSME (Ours) | **0.42** | **0.19** | **0.39** |

Table 1: **Subgroup-specific performance estimation ($n_l = 20$, $n_u = 1000$).** SSME achieves the lowest rescaled metric estimation error (RMAE), averaging across metrics and subgroups.

1.3×. SSME achieves significantly lower metric estimation error in 8 of the 12 combinations of task, metric, and $n_\ell$, and performs comparably to the next best baseline in 3 of the remaining 4 combinations (Tables S11, S12).

## 5.3   Case study: Evaluating subgroup-specific performance

SSME can be applied to measure performance within demographic subgroups, a task central to assessments of algorithmic fairness [12]. We conduct our analysis in the context of critical outcome prediction on MIMIC-IV, a task with documented prediction disparities [50], and a set of commonly studied demographic groups (age, sex, race/ethnicity). We produce subgroup-specific performance estimates by sampling ground truth labels $y$ according to our estimated $p(y|\mathbf{s})$ for only those $\mathbf{s}$ observed among a given demographic group. We perform this analysis with respect to demographic groups based on age, sex, and race/ethnicity. When there is no labeled data for a given subgroup, *Labeled* estimates subgroup-specific performance as global performance.

We report each method's reduction in estimation error relative to labeled data (averaging over metrics and subgroups) for each demographic category in Table 1, comparing to all baselines that can estimate the four performance metrics we average over. SSME reduces metric estimation error by 5.3× on sex, 2.6× on race, and 2.4× on age relative to labeled data, and to a greater extent than all baselines. SSME outperforms all baselines on all individual metrics except for AUC, for which BC reduces estimation error by 3.2× as compared to 2.2× for SSME. BC is well-suited to estimating AUC because it assumes monotonicity when mapping $s$ to $y$. ; i.e., $P(y = 1|s^{(i)}) \geq P(y = 1|s^{(j)})$ when

$s^{(i)} \geq s^{(j)}$. When the classifiers in question have high AUCs — as the critical outcome classifiers do — this is a useful assumption.

## 6  Related Work

Our work unifies ideas from two literatures: methods to 1) evaluate a single classifier and 2) evaluate the accuracy of multiple discrete annotators in the presence of labeled and unlabeled data.

**Classifier evaluation** is a long-standing problem in machine learning. Early work on the use of unlabeled data for classifier evaluation made progress via assumptions about the available unlabeled data (for example, assumptions about class priors [6, 23, 47]; conditionally independent features [64]; well-calibrated classifiers [30]; models of covariate shift [13, 45]). SSME avoids such assumptions by using both labeled and unlabeled data to learn a flexible semi-supervised mixture model of classifier predictions. Interest in unsupervised evaluation has surged in the context of large language models, where models are often judged by other models [77, 34]. These approaches, while promising, are tailored to text generation and not applicable to broader classification tasks. A parallel line of work aims to make use of both labeled and unlabeled data to estimate classifier performance. Existing methods operate on individual classifiers and assume specific parametric forms for the prediction distributions [69, 15, 36, 37, 25]. Some works also focus on model selection in semi-supervised settings, though they do not explicitly estimate performance [46]. SSME differs from these works by naturally capitalizing on multiple classifiers, continuous classifier scores, and unlabeled data. As our results show, doing so leads to improved estimates of performance.

**Evaluation of multiple discrete annotators** was first introduced by [18], who proposed a method to estimate ground truth in the presence of multiple potentially noisy discrete annotations. Many subsequent works inherit Dawid-Skene's strong assumption of class-conditional independence of annotator errors [52, 57], including popular approaches in weak supervision [58, 4, 27], where annotators are instead user-provided labeling functions. Such an assumption is plausible in certain contexts, but does not naturally translate to sets of candidate classifiers, whose predictions are likely to be correlated. However, methods from this line of work are designed to estimate the accuracy of *binary* annotations; they do not exploit the continuous probabilities available in multi-classifier evaluation. While some work has made progress towards accommodating continuous predicted probabilities [51, 56], their focus is optimal aggregation, in contrast to our own, which is evaluation. Recent works use a continuous notion of classifier confidence in conjunction with discrete annotations from each annotator [29, 11], but do not use the distribution of classifier scores over *all* classes.

## 7  Discussion

In this paper, we presented Semi-Supervised Model Evaluation (SSME), a method that supplements limited labeled data with *unlabeled data* to more accurately estimate classifier performance. SSME leverages three features of the current machine learning landscape: (i) there are frequently multiple classifiers for the same task, (ii) continuous classifier scores are often available for all classes, and (iii) unlabeled data is often far more plentiful than labeled data. We provide theoretical bounds on performance estimation error that help explain SSME's advantages over prior work. We show that across multiple tasks, architectures, and modalities, SSME substantially outperforms using labeled data alone and standard baselines.

Our results, and their accompanying limitations, suggest several directions for future work. First, each of the metrics we examined (e.g., AUC) evaluate a single classifier. But because SSME estimates the full joint distribution $P(y, \mathbf{s})$, it could be used to measure properties of the classifiers as a set. For instance, recent work has highlighted the importance of measuring *algorithmic monoculture* [41] where all classifiers produce errors on the same instances. Second, our experiments assess settings in which the unlabeled data is sampled from the same distribution as the labeled data. Although this is common — for example, when a random subset of examples is annotated — many real-world scenarios violate this assumption, and extending SSME to settings with such distribution shift represents an important next step [61]. Finally, our current implementation relies on kernel density estimation, which can scale poorly in high-dimensional spaces (e.g., with many classifiers or classes). Exploring more scalable density estimators could further broaden SSME's applicability. More generally, our results indicate that SSME offers a principled and practical framework for accurate performance estimation in the absence of a large labeled dataset.

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

# Appendix

## A    Related work

We detail related work in unsupervised performance estimation here. Works below assume access to *only* unlabeled data; in contrast, SSME learns from both labeled and unlabeled data.

Unsupervised performance estimation involves estimating the performance of a model given only unlabeled data. Methods designed to address this problem often focus on out-of-distribution samples, where labeled data is scarce and model performance is known to degrade. Several works have illustrated strong empirical relationships between out-of-distribution generalization and thresholded classifier confidence [28], dataset characteristics [21, 30], in-distribution classifier accuracy [48], and classifier agreement [52, 57, 5].

Several works have formalized when unsupervised model evaluation is possible [23, 13, 28, 47, 73, 20, 62 **?** ], and propose assumptions under which estimates of performance are recoverable. [23] and [6] assume knowledge of $p(y)$ in the unlabeled sample. [64] assume conditionally-independent subsets of the observed features, inspired by conditional-independence assumptions made in works such as [18]. [30] assume classifier calibration on unlabeled samples. [13] assume a sparse covariate shift model, in which a subset of the features' class-conditional distribution remains constant. [47] illustrate misestimation of $p(y)$ in the unlabeled example, and assume that $p(y)$ out-of-distribution is close to $p(y)$ in-distribution. As [28] highlight, assumptions are necessary to make any claim about the nature of unsupervised model evaluation, and the above methods are a representative sample of assumptions made Finally, there has been a surge of interest in unsupervised performance estimation in the context of large language models [77, 34]. A standard approach here is to use a large language model to adjudicate the quality of text generated by other language models. Methods in this literature are often specific to large language models, while SSME is not.

Our work is also similar, in spirit, to methods that learn to debias classifier predictions on a small set of labeled data and then apply that debiasing procedure to classifier predictions on unlabeled examples. Prediction-powered inference [3] and double machine learning [14] both learn a debiasing procedure to ensure that unlabeled metric estimates (e.g., accuracy) are statistically unbiased. One of the baselines we compare to, AutoEval [11], is built atop prediction-powered inference.

## B    Experimental details

We provide an extensible implementation of SSME at `https://github.com/divyashan/SSME`, along with support for applying SSME to your own setting. Below, we provide additional detail on the experiments reported in the main text.

### B.1    Real datasets and classifier sets

We provide additional detail for the five datasets we use in our work, including ground truth $p(y)$ for each dataset and ground truth metrics for each classifier in the associated classifier set in Table S1 and Table S2. As discussed, each dataset is split into a training split (provided to each classifier as training data), an estimation split (provided to each performance estimation method), and an evaluation split (used to compute ground truth metrics for each classifier). We determine training splits based on prior work. We then split the remaining data in half (randomly, for each run) to produce the estimation and evaluation splits. We then subsample the estimation split to have $n_l$ labeled examples and $n_u$ unlabeled examples. We ensure that the labeled data always includes at least one example from each class. Thus, the estimation split contains $n_l + n_u$ examples in each experiment, and the evaluation split for each task is fixed across runs (exact sample sizes reported below). No performance estimation method sees data from the evaluation split, which is used to evaluate the performance estimates.

1. **MIMIC-IV**: We use three binary classification tasks from MIMIC-IV [38], a large dataset of electronic health records describing 418K patient visits to an emergency department. We focus on three tasks: **hospitalization** (predicting hospital admission based on features available during triage, $p(y = 1) = 0.45$), **critical outcomes** (predicting inpatient mortality or a transfer to the ICU within 12 hours, $p(y = 1) = 0.06$), and **emergency department revisits** (predicting a patient's return to the emergency department within 3 days, $p(y =$

1) $= 0.03$). We split and preprocess data according to prior work [72, 50]. No patient appears in more than one split. For each task, the evaluation split contains 70,439 examples. The classifiers in the associated set differ by function class (logistic regression, decision tree, and multi-layer perceptron) and random seed (0, 1, 2).

2. **Toxicity detection**: The task is to predict presence of toxicity given an online comment, using data from CivilComments [9, 42] where $p(y = 1) = 0.11$. The evaluation split contains 66,891 examples. The classifiers in the associated set differ by training loss (ERM, IRM, and CORAL) and random seed (0, 1, 2).

3. **Biochemical property prediction** The task is to predict presence of a biochemical property based on a molecular graph, using data from the Open Graph Benchmark [33]. We focus on the task of predicting whether a molecule inhibits SARS-CoV virus maturation, where $p(y = 1) = 0.09$. We filter out examples for which *no* label is observed (i.e. the molecule was not screened at all) because it is impossible to evaluate our performance estimates on those examples. Doing so reduces data held-out from training from 43,793 to 28,325 examples. The evaluation split then contains half, or 14,163, of those examples. The classifiers in the associated set differ by training loss (ERM, IRM, and CORAL) and random seed (0, 1, 2).

4. **News classification** The task is to predict one of four news types based on the title and description of an article [75]. The dataset contains 7,600 examples, with 2,550 examples reserved for the evaluation split. The classes are balanced. Classifiers differ by the base LLM used to perform news classification. The first LLM is Llama-3.2-3B-Instruct, and we obtain probabilities over all classes through zero-shot prompting (i.e. proving one example at a time, accompanied by options) using code provided by [74]. The three remaining LLMs are closed-source models provided by OpenAI: text-embedding-3-small, text-embedding-3-large, and text-embedding-ada-002. We train a classifier atop these embeddings by training a multiclass logistic regression on 2,500 embeddings of labeled examples.

5. **Sentence classification** The task is to predict one of three textual entailments from a sentence [70]. The classes are balanced and the evaluation split contains 61,856 examples. Classifiers differ by training loss (ReWeight, ReSample, IRM, and SqrtReWeight) according to [73].

SSME, in contrast to prior work, makes no asumption about the correlation between classifiers because any assumption is unlikely to hold in practice. The average correlation between classifiers in our sets for each binary task is 0.53, 0.85, 0.93, 0.81, 0.77 (for ED revisit, critical outcome, hospitalization, toxicity, and SARS-COV inhibition prediction respectively). This range of values reflects natural correlation between classifiers in practice, since each of our models is either an off-the-shelf classifier or trained using publicly available code.

### B.2 Baselines

For baselines that require discrete predictions (i.e. Dawid-Skene and AutoEval), we discretize classifier scores by assigning a class according to the maximum classifier score across classes. We expand on our implementation of each baseline below.

- *Labeled*: When estimating performance over the whole dataset, we compare the classifier scores to the ground truth labels within the labeled sample. However, when estimating subgroup-specific performance, it is often the case that there are no labeled examples for a given subgroup. In these instances, *Labeled* reverts to estimating subgroup-specific performance as performance over all labeled examples.

- *Pseudo-Labeling (PL)*: We train a logistic regression with the default parameters associated with the scikit-learn implementation [54]. Experiments with alternative function classes (e.g. a KNN) revealed no significant differences in performance.

- *Bayesian-Calibration (BC)*: Bayesian-Calibration operates on each classifier individually. We make use of the implementation made available by [37]. Extending the proposed approach to multi-class tasks is not straightforward, so we compare to *Bayesian-Calibration* only on binary tasks. We compare to *Bayesian-Calibration* on binary tasks, as it does not extend naturally to multiclass settings.

- *SPE*: SPE [69] ise a semi-supervised single classifier evaluation method that relies on parametric assumptions about the distribution of classifier scores. They find that the truncated

| Dataset | Classifier | Acc | ECE | AUC | AUPRC |
|---|---|---|---|---|---|
| Hospital Admission | DT-RandomForest-seed1 | 74.2 | 1.5 | 81.5 | 76.0 |
| | MLP-ERM-seed2 | 74.4 | 1.4 | 81.7 | 76.7 |
| | MLP-ERM-seed1 | 74.4 | 1.9 | 81.9 | 77.0 |
| | MLP-ERM-seed0 | 74.5 | 2.4 | 82.0 | 77.0 |
| | LR-LBFGS-seed2 | 73.3 | 4.0 | 80.7 | 75.5 |
| | LR-LBFGS-seed1 | 73.3 | 4.0 | 80.7 | 75.5 |
| | LR-LBFGS-seed0 | 73.4 | 2.9 | 81.0 | 75.7 |
| | DT-RandomForest-seed2 | 74.3 | 1.6 | 81.5 | 76.1 |
| | DT-RandomForest-seed0 | 74.1 | 1.5 | 81.5 | 76.1 |
| Critical Outcome | MLP-ERM-seed2 | 93.9 | 0.9 | 87.9 | 38.6 |
| | MLP-ERM-seed1 | 93.9 | 0.8 | 88.1 | 39.0 |
| | LR-LBFGS-seed2 | 93.6 | 1.2 | 87.6 | 34.2 |
| | MLP-ERM-seed0 | 93.9 | 0.5 | 87.5 | 37.8 |
| | LR-LBFGS-seed0 | 93.6 | 1.2 | 87.6 | 34.1 |
| | DT-RandomForest-seed2 | 94.0 | 0.3 | 87.2 | 38.2 |
| | DT-RandomForest-seed1 | 94.0 | 0.4 | 87.4 | 38.3 |
| | DT-RandomForest-seed0 | 94.0 | 0.4 | 87.4 | 38.3 |
| | LR-LBFGS-seed1 | 93.6 | 1.2 | 87.6 | 34.2 |
| ED Revisit | DT-RandomForest-seed0 | 97.7 | 1.8 | 54.9 | 2.7 |
| | DT-RandomForest-seed1 | 97.7 | 1.7 | 55.3 | 2.7 |
| | DT-RandomForest-seed2 | 97.7 | 1.8 | 54.9 | 2.7 |
| | LR-LBFGS-seed0 | 97.7 | 0.4 | 59.3 | 3.0 |
| | LR-LBFGS-seed2 | 97.7 | 0.4 | 59.1 | 3.0 |
| | MLP-ERM-seed0 | 97.7 | 0.3 | 59.8 | 3.1 |
| | MLP-ERM-seed1 | 97.7 | 0.3 | 59.8 | 3.1 |
| | MLP-ERM-seed2 | 97.7 | 0.5 | 57.9 | 3.0 |
| | LR-LBFGS-seed1 | 97.7 | 0.4 | 59.1 | 3.0 |
| Toxicity Detection | distilbert-CORAL-seed0 | 88.3 | 6.0 | 86.2 | 40.0 |
| | distilbert-IRM-seed2 | 88.7 | 10.2 | 91.9 | 65.5 |
| | distilbert-IRM-seed1 | 89.0 | 9.8 | 91.0 | 66.5 |
| | distilbert-IRM-seed0 | 88.1 | 10.6 | 91.6 | 65.9 |
| | distilbert-ERM-seed2 | 92.1 | 4.9 | 94.1 | 73.3 |
| | distilbert-ERM-seed1 | 92.2 | 6.2 | 93.8 | 72.3 |
| | distilbert-ERM-seed0 | 92.2 | 6.1 | 93.8 | 72.2 |
| Molecule Property 60 | gin-virtual-CORAL-seed1 | 92.8 | 5.2 | 90.1 | 61.9 |
| | gin-virtual-CORAL-seed2 | 92.8 | 5.2 | 90.1 | 61.9 |
| | gin-virtual-ERM-seed0 | 94.6 | 1.2 | 94.5 | 73.5 |
| | gin-virtual-ERM-seed1 | 92.4 | 5.6 | 90.7 | 61.1 |
| | gin-virtual-ERM-seed2 | 92.8 | 5.2 | 90.1 | 61.9 |
| | gin-virtual-IRM-seed0 | 93.2 | 1.8 | 90.2 | 58.4 |
| | gin-virtual-IRM-seed1 | 91.1 | 5.2 | 83.8 | 43.8 |
| | gin-virtual-IRM-seed2 | 91.1 | 5.7 | 82.8 | 44.7 |

Table S1: **Ground truth classifier metrics on binary tasks.** We report ground truth performance for classifiers in the sets associated with each binary task. Each classifier name begins with the architecture (e.g. DT represents DecisionTree), the loss or training procedure (e.g. ERM or IRM), and then the seed. Note that the equivalent accuracies on ED Revisit are a byproduct of both the low class prevalence and the poor classifiers.

| dataset | model | Acc | ECE |
|---|---|---|---|
| AG News | llama-3.2-3B-Instruct | 85.6 | 8.6 |
| | text-embedding-3-large | 90.1 | 8.8 |
| | text-embedding-3-small | 89.8 | 9.1 |
| | text-embedding-ada-002 | 89.0 | 9.3 |
| MultiNLI | distilbert-IRM | 64.8 | 6.1 |
| | distilbert-ReSample | 81.4 | 8.2 |
| | distilbert-ReWeight | 80.9 | 7.4 |
| | distilbert-SqrtReWeight | 81.4 | 9.2 |

Table S2: **Ground truth classifier metrics on multiclass tasks.** We report ground truth performance for classifiers in the sets associated with each multiclass task. Each of the LLMs fine-tuned for AG News are sentence transformers, while the MultiNLI classifiers all use DistilBERT [63] as the base architecture.

Normal distribution demonstrates reasonable fit across datasets; accordingly, we implement SPE as a mixture of truncated Normal distributions fit to each classifier's predictions individually.

- *Dawid-Skene*: We implement Dawid-Skene with a tolerance of 1e-5 and a maximum number of EM iterations of 100 (the default parameters), using the following public implementation: `https://github.com/dallascard/dawid_skene`. Dawid-Skene accepts discrete predictions, so we discretize classifier predictions using thresholding the predicted class probability at $\frac{1}{K}$.
- *Majority-Vote*: We implement Majority-Vote as the accuracy-weighted average of discrete predictions made by each classifier. We discretize predictions by thresholding predicted probabilities at $\frac{1}{K}$. We weight each classifier in proportion to its accuracy on the available labeled data.
- *Active-Testing*: We implement Active-Testing, where the method selects a fixed number of examples to label out of a pool of unlabeled examples, according to the approach proposed by [43]. We select examples according to the acquisition strategy for estimating accuracy, a metric for which a public implementation is available, and limit our comparison to this metric.
- *AutoEval*: We implement AutoEval using an implementation made available by the authors [11]. The implementation, to the best of our knowledge, only supports accuracy estimation across a set of classifiers, so we limit our comparison to this metric. We compare to *AutoEval* on accuracy estimation, as additional metrics are not supported by the public implementation.

### B.3 Computing effective sample size

In order to compute effective sample size, we produce 50 samples of labeled data for each increment of 5 between 10 labeled examples and 1000. We then compute the mean absolute metric estimation error of using labeled data alone, across all runs. The effective sample size of a given semi-supervised evaluation method is thus the amount of labeled data which achieves the most similar mean absolute metric estimation error.

## C Proof of Theorem 1

We derive a high-probability error bound on performance estimates based on prior results in semi-supervised mixture models [67]. The proof consists of two steps: we first bound the error in estimation of component separation. We then use the relationship between component separation and classifier performance to bound the error in estimating classifier performance.

**Model of $y$ and s.** We consider a stylized binary classification setting drawn from previous work [67] where classifier scores s are drawn from a balanced mixture of two Gaussians. For data in the positive class, classifier scores are drawn from a multivariate Gaussian The Gaussian assumption holds when the distribution of classifier logits follows a normal distribution. $\mathcal{N}(\mathbf{c}, I)$, where $\mathbf{c} \in \mathbb{R}^d$. For data in the negative class, classifier scores are drawn from a multivariate Gaussian $\mathcal{N}(0, I)$. Let $n_u$ refer to the number of unlabeled examples, $n_\ell$ the number of labeled examples, and $d$ the number of classifiers. Additional classifiers increase the dimensionality of the multivariate Gaussian.

**Estimator** We reason about SSME's performance using a simplified semi-supervised learning algorithm. The estimator, UL+, first uses the unlabeled data to identify decision boundaries, and then uses the labeled data to assign decision boundaries to classes. As the authors note, this is not the most efficient use of labeled data. We expect that using the labeled data in the expectation-maximization procedure – as SSME does – to weakly outperform UL+ because it uses labeled data when learning the decision boundaries. For additional detail on the estimator, please refer to Section 2.3 of [67].

**Theorem 1** Let $\widehat{AUC}_k$, $\widehat{ACC}_k$ be estimators for $AUC_k$, $ACC_k$, estimated using UL+, and $\mathbf{c}_k$ be the L2 distance between components along dimension $k$. Assume $n_u \gtrsim \max\left\{d, \frac{d}{\mathbf{c}^4}, \frac{\log(n_u)}{\min\{\mathbf{c}^2, \mathbf{c}^4\}}, \frac{d\log(dn_u)}{\mathbf{c}^6}\right\}$ and $d \geq 2$. With probability $1 - p$, errors in the estimated AUC and accuracy of classifier $k$ differ by at most:

$$|AUC_k - \widehat{AUC}_k| \leq \quad \Phi\left(\frac{\mathbf{c}_k}{\sqrt{2}}\right) - \Phi\left(\frac{\mathbf{c}_k - \epsilon_{\mathbf{c}}}{\sqrt{2}}\right) \tag{5}$$

$$|ACC_k - \widehat{ACC}_k| \leq \quad \Phi\left(\frac{\mathbf{c}_k}{2}\right) - \Phi\left(\frac{\mathbf{c}_k - \epsilon_{\mathbf{c}}}{2}\right) \tag{6}$$

where $\epsilon_{\mathbf{c}} \lesssim \frac{1}{p}\left(\sqrt{\frac{d}{||\mathbf{c}||^2 n_u}} + ||\mathbf{c}||e^{-\frac{1}{2}n_l||\mathbf{c}||^2\left(1 - \frac{C_0}{||\mathbf{c}||^2}\sqrt{\frac{d\log(n_u)}{||\mathbf{c}||^2 n_u}}\right)^2}\right)$ and $C_0$ is a universal constant.

**1. Bound error in separation estimation as a function of $n_l$, $n_u$, $d$, and separation c.**  Prior work [67] has established bounds on the error in mean estimation $\epsilon_\mu$ in the context of a semi-supervised learning algorithm which first uses the unlabeled data to generate decision boundaries, and uses the labeled data to assign labels to regions. The authors prove mean estimation error can be bounded by:

$$\mathbb{E}\left[\epsilon_\mu\right] \lesssim \sqrt{\frac{d}{||\mathbf{c}||^2 n_u}} + ||\mathbf{c}||e^{-\frac{1}{2}n_l||\mathbf{c}||^2\left(1 - \frac{C_0}{||\mathbf{c}||^2}\sqrt{\frac{d\log(n_u)}{||\mathbf{c}||^2 n_u}}\right)^2}.$$

where $C_0$ is some universal constant and not equivalent to $||\mathbf{c}||$. We can convert this expectation bound into a high-probability bound using Markov's inequality:

$$P\left(\epsilon_\mu \gtrsim \frac{1}{p}\left(\sqrt{\frac{d}{||\mathbf{c}||^2 n_u}} + ||\mathbf{c}||e^{-\frac{1}{2}n_l||\mathbf{c}||^2\left(1 - \frac{C_0}{||\mathbf{c}||^2}\sqrt{\frac{d\log(n_u)}{||\mathbf{c}||^2 n_u}}\right)^2}\right)\right) \leq p.$$

Because we pin $\mu_0 = 0$, error in mean estimation is equal to error in separation estimation in this setting, i.e. $\epsilon_\mu = \epsilon_{\mathbf{c}}$. So we can state that, with probability $1 - p$, the mean estimation error obeys the following inequality:

$$\epsilon_c \leq \frac{1}{p}\left(\sqrt{\frac{d}{||\mathbf{c}||^2 n_u}} + ||\mathbf{c}||e^{-\frac{1}{2}n_l||\mathbf{c}||^2\left(1 - \frac{C_0}{||\mathbf{c}||^2}\sqrt{\frac{d\log(n_u)}{||\mathbf{c}||^2 n_u}}\right)^2}\right)$$

**2. Bound error in performance estimation.**  The AUC of the optimal linear classifier (separating classifier predictions for the negative class from classifier predictions in the positive class) is a function of the Mahalanobis distance **c**, equivalent to the L2 distance in this setting, as prior work has shown [55]:

$$\text{AUC} = \Phi\left(||\mathbf{c}||/\sqrt{2}\right),$$

where $\Phi$ is the cumulative distribution function of the standard normal distribution. Similarly, the accuracy of the optimal linear classifier is:

$$\text{ACC} = \Phi(||\mathbf{c}||/2)$$

Above, we derived a bound on the error of separation estimation, $\epsilon_c$. Now, we map the error in separation estimation to errors in performance estimation (AUC and accuracy). We can state that the following inequality holds:

$$Pr\left(||\mathbf{c}|| - \epsilon_{\mathbf{c}} \leq ||\hat{\mathbf{c}}|| \leq ||\mathbf{c}|| + \epsilon_{\mathbf{c}}\right) \geq 1 - p$$

Thus:

$$\Phi\left(\frac{||\mathbf{c}|| - \epsilon_{\mathbf{c}}}{\sqrt{2}}\right) \leq \Phi\left(\frac{||\hat{\mathbf{c}}||}{\sqrt{2}}\right) \leq \Phi\left(\frac{||\mathbf{c}|| + \epsilon_{\mathbf{c}}}{\sqrt{2}}\right) \tag{7}$$

$$\Phi\left(\frac{||\mathbf{c}|| - \epsilon_{\mathbf{c}}}{\sqrt{2}}\right) \leq \widehat{AUC} \leq \Phi\left(\frac{||\mathbf{c}|| + \epsilon_{\mathbf{c}}}{\sqrt{2}}\right) \tag{8}$$

$$\Phi\left(\frac{||\mathbf{c}|| - \epsilon_{\mathbf{c}}}{\sqrt{2}}\right) - \Phi\left(\frac{||\mathbf{c}||}{\sqrt{2}}\right) \leq \widehat{AUC} - AUC \leq \Phi\left(\frac{||\mathbf{c}|| + \epsilon_{\mathbf{c}}}{\sqrt{2}}\right) - \Phi\left(\frac{||\mathbf{c}||}{\sqrt{2}}\right) \tag{9}$$

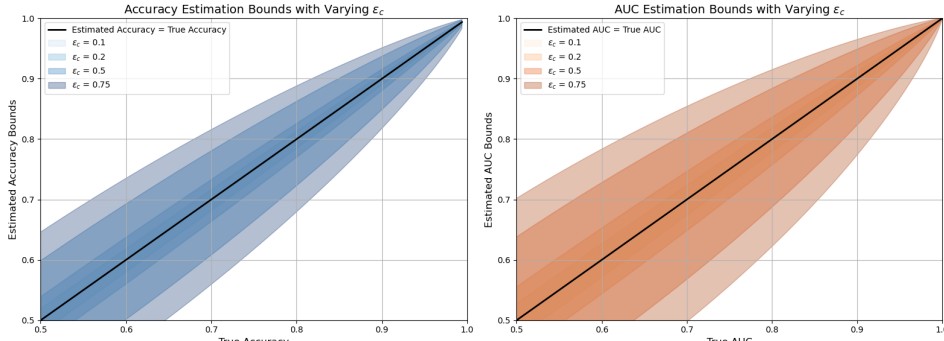

Figure S1: **Estimation error based on original classifier set performance.** We plot estimated classifier performance as a function of original classifier performance for various values of $\epsilon_{\mathbf{c}}$, for accuracy (left) and AUC (right). The less accurate the classifier set, the larger the error bands.

Given that we assume the classifiers are better than random (i.e. $||\mathbf{c}|| > 0$) the term on the far left of the inequality is larger in magnitude than the term on the far right, so we can simplify as:

$$|\widehat{\text{AUC}} - \text{AUC}| \leq \max\left(\Phi\big(\frac{||\mathbf{c}|| + \epsilon_{\mathbf{c}}}{\sqrt{2}}\big) - \Phi\big(\frac{||\mathbf{c}||}{\sqrt{2}}\big), \Phi\big(\frac{||\mathbf{c}|| - \epsilon_{\mathbf{c}}}{\sqrt{2}}\big) - \Phi\big(\frac{||\mathbf{c}||}{\sqrt{2}}\big)\right) \quad (10)$$

$$|\widehat{\text{AUC}} - \text{AUC}| \leq |\Phi\big(\frac{||\mathbf{c}|| - \epsilon_{\mathbf{c}}}{\sqrt{2}}\big) - \Phi\big(\frac{||\mathbf{c}||}{\sqrt{2}}\big)| \quad (11)$$

Finally, error in estimated separation is necessarily larger than the error in any one dimension, allowing us to state the inequality for each dimension of the mixture. We can also follow the same logic above to derive a bound on error in estimated accuracy. We can state:

$$|\widehat{AUC}_k - AUC_k| \leq |\Phi\big(\frac{\mathbf{c}_k}{\sqrt{2}}\big) - \Phi\big(\frac{\mathbf{c}_k - \epsilon_{\mathbf{c}}}{\sqrt{2}}\big)| \quad (12)$$

$$|\widehat{ACC}_k - ACC_k| \leq |\Phi\big(\frac{\mathbf{c}_k}{2}\big) - \Phi\big(\frac{\mathbf{c}_k - \epsilon_{\mathbf{c}}}{2}\big)| \quad (13)$$

where the following holds with probability $1 - p$:

$$\epsilon_{\mathbf{c}} \lesssim \frac{1}{p}\left(\sqrt{\frac{d}{||\mathbf{c}||^2 n_u}} + ||\mathbf{c}||e^{-\frac{1}{2}n_l||\mathbf{c}||^2\left(1 - \frac{C_0}{||\mathbf{c}||^2}\sqrt{\frac{d\log(n_u)}{||\mathbf{c}||^2 n_u}}\right)^2}\right)$$

We plot bounds in estimated classifier performance as a function of $\epsilon_{\mathbf{c}}$ in Fig. S1.

## C.1 Impact of additional classifiers

Increasing the number of classifiers impacts the bound in two ways. First, an additional classifier increases the dimensionality $d$ of the mixture model, thereby creating a looser bound. Second, an additional classifier increases the separation between components $||\mathbf{c}||$, because each classifier is assumed to be independent from one another. If the marginal increase in separation outweighs the marginal increase to dimensionality, the bound on $\epsilon_{\mathbf{c}}$ tightens. Specifically, let's say that the separation of components for the new classifier (i.e. $\mathbf{c}_{k+1}$) is equal to $\delta$. Separation $||\mathbf{c}||$ increases to $\sqrt{||\mathbf{c}||^2 + \delta^2}$. Dimensionality increases by 1, i.e. $d + 1$. When $\sqrt{||\mathbf{c}||^2 + \delta^2}$ exceeds $\sqrt{d + 1}$, the first term in the expression decreases. Specifically:

The norm of the augmented vector $\mathbf{c}' = [\mathbf{c}; \delta]$ exceeds $\sqrt{d + 1}$ if and only if

$$\delta > \sqrt{d + 1 - ||\mathbf{c}||^2}.$$

If $\|\mathbf{c}\| > \sqrt{d+1}$, then any $\delta > 0$ will result in $\|\mathbf{c}'\| > \sqrt{d+1}$. Otherwise, $\delta$ must be large enough to compensate for the square root of the difference between the new dimensionality $(d+1)$ and the squared norm ($\|\mathbf{c}\|^2$). At an extreme, if the new classifier is perfectly correlated with another, it increases dimensionality at no benefit to separation between components, resulting in a looser bound.

## D  Supplementary results

### D.1  Results on synthetic data

We corroborate theoretical findings from Section 3.2 in synthetic experiments, including settings where the assumptions do not hold. We draw classifier scores for positive examples from distributions centered on a vector $\mathbf{c}$, where entries of $\mathbf{c}$ are drawn from $N(c, .2)$, enforcing a minimum entry of 0.01 (to ensure that each classifier exhibits performance better than random). We consider $n_u \in 50, 100, 250, 500, 1000, 2000$, $d \in 2, 4, 6, 8, 10$, and $\|c\| \in 0.75, 1.0, 1.25, 1.5$, and sample 50 datasets for each setting. We provide 20 labeled examples to SSME across all synthetic experiments.

The three theoretical findings we make (using a simplified semi-supervised learning algorithm and a Gaussian assumption) hold in our empirical setting. Our synthetic setting differs from our theoretical setting in two key respects: we use SSME to infer performance and we simulate violations of the Gaussian assumption. Our theoretical findings hold in this more realistic setting; Figure S2 plots results.

### D.2  Results reporting mean absolute error

In the main text, we evaluate our method and all baselines using 20 labeled examples and 1000 unlabeled examples and report *rescaled* mean absolute error across metrics and tasks. Here, we supplement those results by reporting mean absolute error across each task and metric and expanding $n_l$ to include 50 and 100. The number of unlabeled examples remains the same (1000) to isolate the effect of additional labeled data.

Tables S3, S4, S5, and S6 report results on each binary task for accuracy, ECE, AUC, and AUPRC, respectively. Three high-level findings emerge. First, SSME-KDE achieves the lowest mean absolute error (averaging across tasks and amounts of labeled data). Second, SSME-KDE consistently outperforms the ablated version of SSME, fit to a single model at a time (SSME-KDE-M). And finally, SSME-KDE is able to produce performance estimates that are quite close, in absolute terms, to ground truth. For example, when given 20 labeled examples and 1000 unlabeled examples, SSME-KDE estimates accuracy within at most 2.5 percentage points of ground truth accuracy (across tasks).

Tables S11 and S12 report our results on the multiclass tasks, for accuracy and ECE respectively. Note that we exclude Bayesian-Calibration from multiclass comparisons because the method does not natively support multiclass recalibration. We also omit AutoEval from Table S12 because the implementation of expected calibration error within the framework is not straightforward.

### D.3  Results reporting absolute performance estimates

Results thus far have reported aggregate errors in performance estimates across classifiers in the set. Here, we include results on a per-classifier basis in the context of toxicity detection, the task for which we have the largest variability in classifier quality (Tables S7, S8, S9, S10). The tables illustrate how SSME's performance manifests on a per-classifier basis, often producing more accurate estimates than the baselines on the classifiers with lowest performance. The tables also make evident that SSME's improvement in performance estimation can be attributed to a significant reduction in the variance of performance estimates across different data splits.

### D.4  Results reporting robustness to kernel choice

We find that SSME's performance estimates are stable across three kernel choices: a Gaussian kernel, an Epachenikov kernel, and an exponential kernel. Table S13 reports our results in the context of toxicity detection and accuracy, but results generalize across tasks and metrics.

| Dataset | $n_\ell$ | $n_u$ | Labeled | Majority-Vote | PL | Dawid-Skene | SPE | BC | AutoEval | Active-Testing | SSME-M | SSME (Ours) |
|---|---|---|---|---|---|---|---|---|---|---|---|---|
| Critical Outcome | 20 | 1000 | 5.19 ± 1.07 | 4.01 ± 0.13 | 4.12 ± 1.07 | 4.36 ± 0.09 | 12.73 ± 5.64 | 2.80 ± 0.62 | 4.78 ± 0.93 | 5.17 ± 0.32 | 1.70 ± 0.27 | **0.67 ± 0.13** |
| | 50 | 1000 | 2.90 ± 0.59 | 4.21 ± 0.12 | 3.06 ± 0.64 | 4.07 ± 0.11 | 16.79 ± 9.66 | 2.07 ± 0.36 | 3.01 ± 0.65 | 5.61 ± 0.40 | 1.65 ± 0.25 | **0.78 ± 0.13** |
| | 100 | 1000 | 2.09 ± 0.41 | 4.10 ± 0.14 | 1.58 ± 0.30 | 3.87 ± 0.11 | 8.61 ± 8.84 | 1.18 ± 0.21 | 2.00 ± 0.32 | 5.48 ± 0.35 | 1.30 ± 0.20 | **0.77 ± 0.13** |
| ED Revisit | 20 | 1000 | 5.11 ± 0.98 | 2.15 ± 0.02 | 5.13 ± 0.89 | 4.02 ± 0.78 | 14.70 ± 10.59 | 4.36 ± 0.77 | 4.70 ± 0.92 | 2.83 ± 0.20 | 1.64 ± 0.34 | **0.45 ± 0.10** |
| | 50 | 1000 | 2.02 ± 0.58 | 2.10 ± 0.03 | 2.73 ± 0.62 | 2.74 ± 0.61 | 17.96 ± 7.41 | 2.47 ± 0.57 | 1.95 ± 0.57 | 3.03 ± 0.26 | 1.46 ± 0.27 | **0.53 ± 0.11** |
| | 100 | 1000 | 1.43 ± 0.32 | 2.00 ± 0.05 | 1.54 ± 0.34 | 1.51 ± 0.33 | 11.69 ± 5.13 | 1.43 ± 0.31 | 1.42 ± 0.29 | 2.64 ± 0.16 | 1.18 ± 0.25 | **0.57 ± 0.11** |
| Hospital Admission | 20 | 1000 | 7.32 ± 1.25 | 16.32 ± 0.66 | 6.86 ± 1.19 | 19.55 ± 0.15 | 14.27 ± 1.76 | 2.48 ± 0.44 | 7.19 ± 1.03 | 9.33 ± 0.84 | 3.29 ± 0.47 | **1.88 ± 0.29** |
| | 50 | 1000 | 5.40 ± 0.83 | 16.37 ± 0.62 | 3.99 ± 0.82 | 18.78 ± 0.16 | 15.68 ± 0.69 | 2.14 ± 0.36 | 5.23 ± 0.68 | 9.25 ± 0.64 | 3.17 ± 0.51 | **1.95 ± 0.27** |
| | 100 | 1000 | 3.64 ± 0.55 | 15.34 ± 0.66 | 3.01 ± 0.53 | 17.81 ± 0.18 | 14.32 ± 0.95 | 2.42 ± 0.33 | 4.01 ± 0.55 | 9.24 ± 0.80 | 3.06 ± 0.45 | **1.51 ± 0.23** |
| SARS-CoV Inhibition | 20 | 1000 | 6.11 ± 0.96 | 2.29 ± 0.42 | 5.95 ± 1.00 | 4.91 ± 0.18 | 11.42 ± 5.06 | **2.25 ± 0.31** | 4.59 ± 0.84 | 5.97 ± 0.47 | 3.06 ± 0.23 | 2.30 ± 0.16 |
| | 50 | 1000 | 3.22 ± 0.57 | 2.46 ± 0.38 | 2.99 ± 0.45 | 4.50 ± 0.17 | 8.04 ± 2.56 | **1.74 ± 0.21** | 2.64 ± 0.42 | 5.74 ± 0.32 | 2.59 ± 0.26 | 2.35 ± 0.10 |
| | 100 | 1000 | 2.04 ± 0.38 | 2.38 ± 0.35 | 2.14 ± 0.31 | 4.01 ± 0.17 | 6.09 ± 4.63 | **1.43 ± 0.19** | 1.94 ± 0.25 | 5.99 ± 0.47 | 1.84 ± 0.24 | 2.36 ± 0.13 |
| Toxicity Detection | 20 | 1000 | 5.95 ± 0.73 | 4.97 ± 0.31 | 5.03 ± 0.81 | 4.82 ± 0.09 | 11.26 ± 5.18 | 5.29 ± 0.29 | | 7.19 ± 0.44 | 6.71 ± 0.23 | **2.34 ± 0.15** |
| | 50 | 1000 | 4.03 ± 0.68 | 4.79 ± 0.26 | 2.88 ± 0.48 | 4.65 ± 0.08 | 8.75 ± 1.66 | 4.57 ± 0.30 | 3.37 ± 0.41 | 7.26 ± 0.47 | 5.38 ± 0.28 | **2.22 ± 0.13** |
| | 100 | 1000 | 2.43 ± 0.41 | 4.46 ± 0.22 | **1.90 ± 0.31** | 4.46 ± 0.11 | 7.41 ± 0.16 | 3.78 ± 0.26 | 2.34 ± 0.26 | 7.50 ± 0.60 | 3.80 ± 0.32 | 2.14 ± 0.15 |

Table S3: **Mean absolute error in accuracy estimation on binary tasks.** We report mean absolute error (MAE, averaging across classifiers) across five binary classification tasks and different amounts of labeled data. Each entry corresponds to the mean MAE across 50 randomized splits of data, accompanied by the 95% CI (computed as $1.96 \times$ the standard error) in MAE across splits. We bold the best performing method in each row, and underline the next best performing method by mean MAE. SSME-M refers to performance when fitting SSME to a single classifier's scores, instead of modeling the joint distribution of $P(y, \mathbf{s})$.

| Dataset | $n_\ell$ | $n_u$ | Labeled | Majority-Vote | PL | Dawid-Skene | SPE | BC | SSME-M | SSME (Ours) |
|---|---|---|---|---|---|---|---|---|---|---|
| Critical Outcome | 20 | 1000 | 11.01 ± 1.12 | 2.75 ± 0.26 | 6.94 ± 0.64 | 2.61 ± 0.09 | 16.17 ± 6.69 | 3.48 ± 0.76 | 3.17 ± 0.30 | **1.16 ± 0.13** |
| | 50 | 1000 | 6.22 ± 0.62 | 3.68 ± 0.34 | 5.38 ± 0.39 | 2.37 ± 0.09 | 20.91 ± 9.43 | 2.56 ± 0.44 | 3.01 ± 0.26 | **1.13 ± 0.13** |
| | 100 | 1000 | 4.20 ± 0.38 | 3.43 ± 0.28 | 3.63 ± 0.21 | 2.25 ± 0.10 | 12.84 ± 8.93 | 1.69 ± 0.25 | 2.81 ± 0.22 | **1.15 ± 0.10** |
| ED Revisit | 20 | 1000 | 8.37 ± 0.87 | 1.81 ± 0.02 | 4.16 ± 0.82 | 3.25 ± 0.68 | 14.15 ± 10.31 | 3.57 ± 0.77 | 1.88 ± 0.24 | **0.76 ± 0.04** |
| | 50 | 1000 | 4.82 ± 0.48 | 1.77 ± 0.03 | 2.29 ± 0.47 | 2.29 ± 0.46 | 17.12 ± 7.09 | 2.04 ± 0.46 | 1.83 ± 0.19 | **0.73 ± 0.05** |
| | 100 | 1000 | 3.29 ± 0.24 | 1.69 ± 0.04 | 1.36 ± 0.22 | 1.34 ± 0.21 | 10.85 ± 4.82 | 1.16 ± 0.20 | 1.51 ± 0.17 | **0.73 ± 0.06** |
| Hospital Admission | 20 | 1000 | 21.76 ± 1.16 | 21.73 ± 2.75 | 8.10 ± 1.28 | 17.31 ± 0.13 | 16.79 ± 0.58 | 5.12 ± 1.09 | 5.54 ± 0.36 | **1.97 ± 0.13** |
| | 50 | 1000 | 12.74 ± 0.62 | 21.03 ± 2.65 | 5.02 ± 0.68 | 16.60 ± 0.13 | 14.87 ± 0.47 | 3.49 ± 0.57 | 5.20 ± 0.33 | **2.06 ± 0.18** |
| | 100 | 1000 | 8.56 ± 0.38 | 19.97 ± 2.45 | 3.91 ± 0.49 | 15.62 ± 0.13 | 13.39 ± 0.79 | 3.23 ± 0.47 | 5.32 ± 0.37 | **1.70 ± 0.15** |
| SARS-CoV Inhibition | 20 | 1000 | 7.44 ± 0.95 | **1.74 ± 0.38** | 5.96 ± 0.87 | 4.35 ± 0.15 | 11.85 ± 5.63 | 2.24 ± 0.33 | 2.57 ± 0.18 | 3.38 ± 0.13 |
| | 50 | 1000 | 3.66 ± 0.50 | 1.96 ± 0.39 | 3.06 ± 0.35 | 4.08 ± 0.16 | 8.96 ± 2.74 | **1.73 ± 0.26** | 2.27 ± 0.20 | 3.41 ± 0.11 |
| | 100 | 1000 | 2.18 ± 0.32 | 1.83 ± 0.37 | 2.36 ± 0.22 | 3.67 ± 0.16 | 7.28 ± 4.71 | **1.35 ± 0.22** | 1.79 ± 0.19 | 3.44 ± 0.13 |
| Toxicity Detection | 20 | 1000 | 5.85 ± 0.80 | 5.08 ± 0.39 | 5.09 ± 0.79 | 4.40 ± 0.09 | 11.23 ± 5.14 | 4.69 ± 0.34 | 5.67 ± 0.19 | **2.35 ± 0.13** |
| | 50 | 1000 | 3.99 ± 0.63 | 4.84 ± 0.33 | 3.04 ± 0.46 | 4.20 ± 0.07 | 7.97 ± 1.76 | 3.97 ± 0.34 | 4.57 ± 0.26 | **2.26 ± 0.12** |
| | 100 | 1000 | 2.37 ± 0.37 | 4.63 ± 0.33 | **1.91 ± 0.27** | 4.10 ± 0.09 | 6.75 ± 0.15 | 3.30 ± 0.25 | 3.43 ± 0.29 | 2.19 ± 0.15 |

Table S4: **Mean absolute error in ECE estimation on binary tasks.**

| Dataset | $n_\ell$ | $n_u$ | Labeled | Majority-Vote | PL | Dawid-Skene | SPE | BC | SSME-M | SSME (Ours) |
|---|---|---|---|---|---|---|---|---|---|---|
| Critical Outcome | 20 | 1000 | 10.09 ± 1.34 | 9.45 ± 0.39 | 31.73 ± 1.10 | 9.39 ± 0.35 | 10.92 ± 4.41 | 2.84 ± 0.25 | 4.72 ± 0.63 | **2.52 ± 0.34** |
| | 50 | 1000 | 7.50 ± 1.28 | 8.06 ± 0.53 | 27.33 ± 1.53 | 8.49 ± 0.40 | 20.24 ± 11.78 | 3.17 ± 0.32 | 5.61 ± 1.28 | **2.39 ± 0.48** |
| | 100 | 1000 | 5.65 ± 0.95 | 7.29 ± 0.61 | 20.43 ± 1.22 | 7.97 ± 0.30 | 13.71 ± 10.26 | 2.70 ± 0.26 | 3.82 ± 0.48 | **2.83 ± 0.80** |
| ED Revisit | 20 | 1000 | 18.48 ± 1.85 | 19.99 ± 2.44 | 7.48 ± 0.20 | 8.27 ± 1.05 | 25.84 ± 6.42 | 7.65 ± 0.15 | 11.89 ± 1.29 | **5.92 ± 0.87** |
| | 50 | 1000 | 17.37 ± 1.98 | 17.93 ± 2.58 | 7.48 ± 0.26 | 7.62 ± 0.28 | 29.83 ± 4.67 | 7.30 ± 0.21 | 11.99 ± 1.21 | **5.09 ± 0.71** |
| | 100 | 1000 | 14.13 ± 1.67 | 14.95 ± 2.29 | 7.06 ± 0.40 | 7.09 ± 0.42 | 27.00 ± 3.59 | 7.47 ± 0.32 | 11.28 ± 1.59 | **5.08 ± 0.77** |
| Hospital Admission | 20 | 1000 | 6.97 ± 1.29 | 16.20 ± 1.00 | 8.94 ± 1.65 | 16.70 ± 0.10 | 15.81 ± 0.33 | 2.67 ± 0.32 | 3.63 ± 0.54 | **2.51 ± 0.38** |
| | 50 | 1000 | 5.08 ± 0.97 | 15.47 ± 0.56 | 5.59 ± 1.20 | 16.18 ± 0.10 | 14.93 ± 0.40 | 2.62 ± 0.46 | 3.18 ± 0.54 | **2.51 ± 0.33** |
| | 100 | 1000 | 3.57 ± 0.71 | 14.78 ± 0.44 | 3.66 ± 0.74 | 15.32 ± 0.12 | 13.95 ± 0.72 | 2.55 ± 0.37 | 3.17 ± 0.44 | **2.02 ± 0.33** |
| SARS-CoV Inhibition | 20 | 1000 | 9.61 ± 2.56 | 5.44 ± 0.60 | 30.92 ± 1.21 | 7.50 ± 0.29 | 13.72 ± 5.03 | **3.07 ± 0.28** | 5.42 ± 0.73 | 3.48 ± 0.44 |
| | 50 | 1000 | 5.84 ± 1.01 | 5.22 ± 0.39 | 22.71 ± 1.19 | 7.06 ± 0.29 | 10.67 ± 2.28 | 3.62 ± 0.27 | 5.02 ± 0.52 | **3.41 ± 0.47** |
| | 100 | 1000 | 3.97 ± 0.55 | 4.95 ± 0.35 | 16.33 ± 0.91 | 6.04 ± 0.32 | 10.34 ± 4.18 | 3.53 ± 0.38 | 4.21 ± 0.53 | **3.46 ± 0.45** |
| Toxicity Detection | 20 | 1000 | 6.71 ± 0.99 | 5.45 ± 0.82 | 17.32 ± 2.08 | 6.20 ± 0.11 | 12.38 ± 5.76 | 5.22 ± 0.16 | 6.05 ± 0.28 | **3.34 ± 0.23** |
| | 50 | 1000 | 4.76 ± 0.91 | 5.15 ± 0.47 | 11.79 ± 1.78 | 5.97 ± 0.09 | 8.23 ± 1.65 | 4.76 ± 0.21 | 4.86 ± 0.29 | **3.15 ± 0.18** |
| | 100 | 1000 | 3.82 ± 0.60 | 4.85 ± 0.20 | 7.54 ± 1.03 | 5.84 ± 0.12 | 7.28 ± 0.22 | 4.25 ± 0.27 | 4.15 ± 0.33 | **3.09 ± 0.22** |

Table S5: **Mean absolute error in AUC estimation on binary tasks.**

| Dataset | $n_\ell$ | $n_u$ | Labeled | Majority-Vote | PL | Dawid-Skene | SPE | BC | SSME-M | SSME (Ours) |
|---|---|---|---|---|---|---|---|---|---|---|
| Critical Outcome | 20 | 1000 | 32.86 ± 5.06 | 36.21 ± 2.09 | 22.98 ± 1.85 | 39.02 ± 1.18 | 45.09 ± 6.81 | 9.29 ± 1.67 | 11.48 ± 1.51 | **6.11 ± 0.73** |
| | 50 | 1000 | 22.81 ± 3.65 | 26.12 ± 2.77 | 20.48 ± 2.23 | 35.64 ± 1.61 | 44.67 ± 7.31 | 9.34 ± 1.43 | 11.98 ± 1.43 | **6.17 ± 0.96** |
| | 100 | 1000 | 15.71 ± 2.44 | 24.01 ± 2.47 | 14.45 ± 2.02 | 33.31 ± 1.20 | 47.35 ± 6.71 | 8.96 ± 1.41 | 11.30 ± 1.67 | **5.77 ± 0.77** |
| ED Revisit | 20 | 1000 | 19.18 ± 3.68 | 4.53 ± 1.59 | 5.14 ± 0.89 | 5.12 ± 1.30 | 16.66 ± 11.60 | 9.14 ± 1.04 | 5.07 ± 0.80 | **1.67 ± 0.29** |
| | 50 | 1000 | 8.85 ± 2.26 | 3.12 ± 0.79 | 2.72 ± 0.62 | 3.04 ± 0.78 | 24.55 ± 10.06 | 6.03 ± 0.82 | 3.79 ± 0.65 | **1.81 ± 0.30** |
| | 100 | 1000 | 6.34 ± 1.54 | 3.23 ± 0.82 | **1.57 ± 0.33** | 1.74 ± 0.34 | 24.75 ± 12.17 | 4.23 ± 0.55 | 3.92 ± 0.62 | 1.82 ± 0.31 |
| Hospital Admission | 20 | 1000 | 9.43 ± 1.62 | 32.72 ± 6.41 | 10.89 ± 2.59 | 21.15 ± 0.16 | 21.13 ± 0.34 | 5.26 ± 1.07 | 4.36 ± 0.44 | **3.47 ± 0.57** |
| | 50 | 1000 | 7.46 ± 1.31 | 31.37 ± 6.28 | 7.91 ± 1.63 | 20.34 ± 0.78 | 19.21 ± 0.80 | 4.43 ± 0.74 | 3.70 ± 0.63 | **3.64 ± 0.60** |
| | 100 | 1000 | 5.51 ± 0.97 | 29.71 ± 5.98 | 4.12 ± 1.02 | 19.17 ± 0.21 | 18.73 ± 0.71 | 3.49 ± 0.60 | 4.00 ± 0.61 | **2.80 ± 0.51** |
| SARS-CoV Inhibition | 20 | 1000 | 22.27 ± 3.03 | 18.75 ± 2.82 | 37.41 ± 2.44 | 16.60 ± 1.08 | 41.22 ± 6.90 | **7.54 ± 0.76** | 13.81 ± 1.53 | 20.51 ± 1.70 |
| | 50 | 1000 | 15.02 ± 2.43 | 14.10 ± 1.05 | 30.29 ± 2.61 | 15.01 ± 1.07 | 34.97 ± 1.49 | **8.40 ± 0.96** | 12.82 ± 1.01 | 21.06 ± 1.51 |
| | 100 | 1000 | 11.53 ± 1.56 | 15.72 ± 2.66 | 20.34 ± 1.79 | 12.61 ± 1.05 | 33.60 ± 2.49 | **8.27 ± 0.89** | 11.01 ± 1.39 | 20.67 ± 1.64 |
| Toxicity Detection | 20 | 1000 | 19.34 ± 2.34 | 29.05 ± 3.31 | 25.12 ± 3.52 | 26.34 ± 0.36 | 37.10 ± 6.70 | 19.94 ± 1.30 | 23.38 ± 0.73 | **10.89 ± 0.87** |
| | 50 | 1000 | 13.78 ± 1.81 | 30.54 ± 3.70 | 20.15 ± 3.48 | 25.24 ± 0.36 | 31.94 ± 1.07 | 16.84 ± 1.58 | 18.90 ± 1.08 | **9.91 ± 0.85** |
| | 100 | 1000 | 10.69 ± 1.71 | 23.87 ± 0.84 | 14.06 ± 2.00 | 24.51 ± 0.47 | 30.94 ± 0.58 | 14.15 ± 1.48 | 14.59 ± 1.26 | **9.88 ± 0.97** |

Table S6: **Mean absolute error in AUPRC estimation on binary tasks.**

| model | Labeled | Majority-Vote | PL | Dawid-Skene | SPE | BC | AutoEval | Active-Testing | SSME (Ours) | Ground Truth |
|---|---|---|---|---|---|---|---|---|---|---|
| distilbert-CORAL | 83.90 ± 14.04 | 83.81 ± 13.35 | 86.15 ± 8.05 | 84.75 ± 2.64 | 73.48 ± 51.60 | 87.82 ± 8.64 | 85.04 ± 10.63 | 87.92 ± 21.20 | 86.49 ± 2.07 | 88.27 ± 0.18 |
| distilbert-ERM | 89.30 ± 12.04 | 91.34 ± 7.53 | 86.92 ± 6.57 | 95.08 ± 1.94 | 84.55 ± 51.41 | 97.13 ± 1.88 | 90.79 ± 11.63 | 89.17 ± 16.14 | 93.59 ± 1.71 | 92.17 ± 0.14 |
| distilbert-ERM-seed1 | 89.10 ± 13.96 | 91.28 ± 7.50 | 87.05 ± 6.54 | 94.86 ± 2.06 | 93.25 ± 10.36 | 97.14 ± 1.97 | 90.75 ± 12.08 | 95.07 ± 12.69 | 93.54 ± 1.63 | 92.17 ± 0.15 |
| distilbert-ERM-seed2 | 89.20 ± 12.42 | 91.63 ± 7.32 | 86.67 ± 6.72 | 95.78 ± 1.84 | 85.49 ± 46.57 | 96.28 ± 2.42 | 90.43 ± 13.38 | 92.84 ± 12.24 | 93.65 ± 1.61 | 92.11 ± 0.15 |
| distilbert-IRM | 86.90 ± 14.93 | 92.67 ± 9.76 | 82.92 ± 6.39 | 95.27 ± 1.29 | 93.48 ± 14.96 | 94.56 ± 4.59 | 88.11 ± 14.15 | 88.10 ± 17.89 | 91.65 ± 1.91 | 88.13 ± 0.18 |
| distilbert-IRM-seed1 | 88.10 ± 14.40 | 92.87 ± 8.98 | 83.86 ± 6.49 | 95.92 ± 1.32 | 95.76 ± 12.55 | 95.41 ± 3.44 | 89.27 ± 13.77 | 89.42 ± 17.90 | 92.26 ± 1.65 | 89.04 ± 0.19 |
| distilbert-IRM-seed2 | 86.80 ± 13.54 | 92.48 ± 9.09 | 83.39 ± 6.35 | 95.55 ± 1.35 | 86.76 ± 52.55 | 95.34 ± 4.27 | 87.78 ± 13.98 | 89.19 ± 18.15 | 92.08 ± 1.78 | 88.70 ± 0.16 |

Table S7: **Mean absolute error in accuracy estimation per classifier on toxicity detection.** .

| model | Labeled | Majority-Vote | PL | Dawid-Skene | SPE | BC | SSME (Ours) | Ground Truth |
|---|---|---|---|---|---|---|---|---|
| distilbert-CORAL | 13.80 ± 11.38 | 12.60 ± 9.19 | 8.78 ± 7.05 | 10.78 ± 2.12 | 25.34 ± 53.26 | 7.47 ± 8.12 | 8.50 ± 1.87 | 5.98 ± 0.18 |
| distilbert-ERM | 10.37 ± 11.48 | 8.16 ± 7.77 | 11.33 ± 6.44 | 4.37 ± 2.08 | 14.87 ± 49.40 | 1.97 ± 1.97 | 4.96 ± 1.47 | 6.14 ± 0.17 |
| distilbert-ERM-seed1 | 9.91 ± 12.33 | 8.34 ± 7.65 | 11.29 ± 6.35 | 4.59 ± 2.21 | 6.45 ± 10.72 | 1.93 ± 2.09 | 5.13 ± 1.53 | 6.21 ± 0.16 |
| distilbert-ERM-seed2 | 10.02 ± 11.80 | 7.70 ± 7.75 | 10.96 ± 6.53 | 3.38 ± 1.93 | 14.36 ± 46.44 | 2.32 ± 2.47 | 3.99 ± 1.39 | 4.94 ± 0.19 |
| distilbert-IRM | 12.59 ± 13.50 | 6.52 ± 9.97 | 15.77 ± 6.04 | 3.69 ± 1.31 | 6.69 ± 14.92 | 4.61 ± 5.14 | 6.85 ± 1.72 | 10.61 ± 0.16 |
| distilbert-IRM-seed1 | 11.93 ± 13.31 | 6.35 ± 9.18 | 15.03 ± 6.38 | 2.98 ± 1.24 | 4.18 ± 12.63 | 3.88 ± 3.89 | 6.38 ± 1.83 | 9.78 ± 0.21 |
| distilbert-IRM-seed2 | 12.06 ± 12.00 | 6.71 ± 9.42 | 15.55 ± 6.16 | 3.33 ± 1.46 | 13.01 ± 51.77 | 3.87 ± 4.60 | 6.71 ± 1.74 | 10.18 ± 0.18 |

Table S8: **Mean absolute error in ECE estimation per classifier on toxicity detection.**

| model | Labeled | Majority-Vote | PL | Dawid-Skene | SPE | BC | SSME (Ours) | Ground Truth |
|---|---|---|---|---|---|---|---|---|
| distilbert-CORAL | 85.19 ± 27.74 | 90.53 ± 8.91 | 72.20 ± 13.51 | 94.89 ± 2.31 | 79.61 ± 72.95 | 84.22 ± 6.84 | 91.38 ± 3.14 | 86.23 ± 0.33 |
| distilbert-ERM | 91.80 ± 13.93 | 96.36 ± 8.70 | 74.90 ± 15.08 | 98.64 ± 0.65 | 90.62 ± 57.77 | 98.52 ± 1.84 | 95.97 ± 1.50 | 93.77 ± 0.22 |
| distilbert-ERM-seed1 | 92.18 ± 14.63 | 96.36 ± 8.76 | 74.92 ± 15.05 | 98.58 ± 0.59 | 99.70 ± 0.58 | 98.30 ± 1.86 | 95.96 ± 1.39 | 93.75 ± 0.20 |
| distilbert-ERM-seed2 | 93.32 ± 11.62 | 96.55 ± 8.86 | 75.03 ± 15.17 | 98.89 ± 0.62 | 90.81 ± 57.23 | 98.07 ± 2.20 | 96.16 ± 1.33 | 94.08 ± 0.19 |
| distilbert-IRM | 91.46 ± 17.13 | 95.52 ± 13.36 | 74.69 ± 14.67 | 98.28 ± 1.17 | 99.06 ± 3.55 | 98.05 ± 1.88 | 95.38 ± 2.22 | 91.57 ± 0.25 |
| distilbert-IRM-seed1 | 90.75 ± 20.55 | 95.25 ± 12.97 | 74.58 ± 14.85 | 97.97 ± 1.60 | 99.52 ± 1.60 | 98.07 ± 2.00 | 95.18 ± 2.19 | 91.00 ± 0.31 |
| distilbert-IRM-seed2 | 91.89 ± 15.94 | 95.37 ± 13.62 | 74.68 ± 14.90 | 98.39 ± 1.02 | 90.44 ± 57.04 | 98.33 ± 1.84 | 95.59 ± 1.79 | 91.86 ± 0.23 |

Table S9: **Mean absolute error in AUC estimation per classifier on toxicity detection.**

| model | Labeled | Majority-Vote | PL | Dawid-Skene | SPE | BC | SSME (Ours) | Ground Truth |
|---|---|---|---|---|---|---|---|---|
| distilbert-CORAL | 60.78 ± 47.30 | 67.66 ± 48.12 | 31.91 ± 19.96 | 76.38 ± 9.81 | 71.94 ± 75.73 | 50.86 ± 21.03 | 60.27 ± 10.78 | 40.00 ± 0.70 |
| distilbert-ERM | 77.25 ± 38.26 | 82.78 ± 55.03 | 42.28 ± 26.91 | 94.59 ± 2.39 | 79.50 ± 71.95 | 92.51 ± 8.47 | 79.63 ± 7.22 | 72.19 ± 0.73 |
| distilbert-ERM-seed1 | 79.38 ± 36.73 | 83.34 ± 53.48 | 42.27 ± 26.76 | 94.37 ± 2.25 | 83.36 ± 69.87 | 91.54 ± 9.68 | 79.78 ± 6.78 | 72.30 ± 0.68 |
| distilbert-ERM-seed2 | 79.11 ± 35.38 | 83.41 ± 55.47 | 42.47 ± 27.03 | 95.49 ± 2.34 | 78.44 ± 73.38 | 90.15 ± 9.69 | 80.06 ± 6.54 | 73.33 ± 0.69 |
| distilbert-IRM | 77.74 ± 38.16 | 81.26 ± 57.61 | 40.79 ± 25.87 | 92.89 ± 2.88 | 81.71 ± 76.90 | 88.23 ± 14.48 | 76.90 ± 8.27 | 65.86 ± 0.78 |
| distilbert-IRM-seed1 | 79.21 ± 39.17 | 81.73 ± 57.61 | 41.23 ± 26.40 | 93.64 ± 3.17 | 82.29 ± 74.37 | 89.58 ± 11.12 | 77.84 ± 7.67 | 66.50 ± 0.86 |
| distilbert-IRM-seed2 | 77.63 ± 39.97 | 80.53 ± 57.24 | 40.65 ± 26.19 | 92.67 ± 3.46 | 77.63 ± 77.91 | 90.34 ± 12.47 | 77.16 ± 7.20 | 65.46 ± 0.79 |

Table S10: **Mean absolute error in AUPRC estimation per classifier on toxicity detection.**

| Dataset | $n_\ell$ | $n_u$ | Labeled | Majority-Vote | PL | Dawid-Skene | AutoEval | Active-Testing | SSME (Ours) |
|---|---|---|---|---|---|---|---|---|---|
| AG News | 20 | 1000 | 5.48 ± 0.89 | 5.13 ± 0.16 | 11.72 ± 1.23 | 6.70 ± 0.13 | 5.59 ± 1.18 | 8.09 ± 0.80 | 3.72 ± 0.19 |
| | 50 | 1000 | 3.92 ± 0.63 | 4.96 ± 0.14 | 5.22 ± 0.78 | 6.51 ± 0.12 | 4.36 ± 0.78 | 8.88 ± 1.40 | 3.54 ± 0.17 |
| | 100 | 1000 | 2.71 ± 0.45 | 4.74 ± 0.19 | 2.69 ± 0.48 | 6.20 ± 0.16 | 2.71 ± 0.59 | 8.77 ± 1.20 | 3.41 ± 0.24 |
| MultiNLI | 20 | 1000 | 7.46 ± 1.07 | 8.30 ± 0.35 | 7.95 ± 1.26 | 11.73 ± 0.15 | 7.20 ± 1.04 | 10.30 ± 1.77 | 1.98 ± 0.24 |
| | 50 | 1000 | 4.42 ± 0.55 | 8.14 ± 0.27 | 3.08 ± 0.62 | 11.41 ± 0.14 | 4.17 ± 0.54 | 11.85 ± 1.70 | 1.90 ± 0.21 |
| | 100 | 1000 | 3.27 ± 0.46 | 7.54 ± 0.30 | 2.47 ± 0.51 | 10.72 ± 0.15 | 3.17 ± 0.44 | 11.63 ± 1.77 | 2.02 ± 0.23 |

Table S11: **Mean absolute error in accuracy estimation on multiclass tasks.**

| Dataset | $n_\ell$ | $n_u$ | Labeled | Majority-Vote | PL | Dawid-Skene | SSME (Ours) |
|---|---|---|---|---|---|---|---|
| AG News | 20 | 1000 | 5.53 ± 0.93 | 4.30 ± 0.13 | 11.85 ± 1.21 | 5.80 ± 0.11 | 3.37 ± 0.17 |
| | 50 | 1000 | 3.88 ± 0.76 | 4.18 ± 0.11 | 5.36 ± 0.77 | 5.70 ± 0.10 | 3.28 ± 0.15 |
| | 100 | 1000 | 2.61 ± 0.49 | 4.07 ± 0.15 | 2.78 ± 0.48 | 5.53 ± 0.13 | 3.13 ± 0.21 |
| MultiNLI | 20 | 1000 | 11.57 ± 1.13 | 3.87 ± 0.18 | 7.84 ± 1.14 | 2.95 ± 0.08 | 2.06 ± 0.24 |
| | 50 | 1000 | 6.14 ± 0.67 | 3.94 ± 0.14 | 3.10 ± 0.62 | 2.92 ± 0.10 | 2.06 ± 0.20 |
| | 100 | 1000 | 4.52 ± 0.51 | 3.82 ± 0.15 | 2.37 ± 0.46 | 3.18 ± 0.08 | 2.19 ± 0.21 |

Table S12: **Mean absolute error in ECE estimation on multiclass tasks.**

| model | SSME-KDE-gaussian | SSME-KDE-epanechnikov | SSME-KDE-exponential |
|---|---|---|---|
| distilbert_CORAL | 86.61 ± 1.3587 | 86.61 ± 1.3586 | 86.60 ± 1.3884 |
| distilbert_ERM | 93.83 ± 1.0066 | 93.83 ± 1.0067 | 93.81 ± 0.9879 |
| distilbert_ERM_seed1 | 93.76 ± 0.9585 | 93.75 ± 0.9584 | 93.73 ± 0.9314 |
| distilbert_ERM_seed2 | 93.77 ± 0.6859 | 93.77 ± 0.6859 | 93.75 ± 0.6643 |
| distilbert_IRM | 91.80 ± 1.0459 | 91.80 ± 1.0460 | 91.78 ± 1.0122 |
| distilbert_IRM_seed1 | 92.37 ± 1.0373 | 92.37 ± 1.0373 | 92.35 ± 1.0252 |
| distilbert_IRM_seed2 | 92.33 ± 0.9593 | 92.33 ± 0.9592 | 92.31 ± 0.9346 |

Table S13: **Robustness of SSME to kernel choice**. Because density estimation can be sensitive to kernel choice, we analyze the stability of SSME's performance estimates across three kernel types. We find no significant imapct of kernel choice on SSME's performance estimates (although different kernels do produce small changes in accuracy estimates). Results hold across metrics and tasks; we report results for each toxicity detection classifier and accuracy for brevity.

| Dataset | $n_\ell$ | $n_u$ | Labeled | Majority-Vote | PL | Dawid-Skene | AutoEval | Active-Testing | SSME (Ours) | SSME-NF |
|---|---|---|---|---|---|---|---|---|---|---|
| AG News | 20 | 1000 | 5.48 ± 0.45 | 5.13 ± 0.08 | 11.72 ± 0.63 | 6.70 ± 0.07 | 5.59 ± 0.60 | 8.09 ± 0.41 | 3.72 ± 0.10 | 3.52 ± 0.11 |
| | 50 | 1000 | 3.92 ± 0.32 | 4.96 ± 0.07 | 5.22 ± 0.40 | 6.51 ± 0.06 | 4.36 ± 0.40 | 8.88 ± 0.72 | 3.54 ± 0.09 | 3.50 ± 0.11 |
| | 100 | 1000 | 2.71 ± 0.23 | 4.74 ± 0.10 | 2.69 ± 0.25 | 6.20 ± 0.08 | 2.71 ± 0.30 | 8.77 ± 0.61 | 3.41 ± 0.12 | 3.40 ± 0.13 |
| MultiNLI | 20 | 1000 | 7.46 ± 0.55 | 8.30 ± 0.18 | 7.95 ± 0.64 | 11.73 ± 0.08 | 7.20 ± 0.53 | 10.30 ± 0.90 | 1.98 ± 0.12 | 3.08 ± 0.09 |
| | 50 | 1000 | 4.42 ± 0.28 | 8.14 ± 0.14 | 3.08 ± 0.32 | 11.41 ± 0.07 | 4.17 ± 0.28 | 11.85 ± 0.87 | 1.90 ± 0.11 | 2.79 ± 0.11 |
| | 100 | 1000 | 3.27 ± 0.23 | 7.54 ± 0.15 | 2.47 ± 0.26 | 10.72 ± 0.08 | 3.17 ± 0.23 | 11.63 ± 0.90 | 2.02 ± 0.12 | 2.52 ± 0.11 |

Table S14: **Mean absolute error in accuracy estimation on multiclass tasks, including NF parametrization.** SSME-NF performs comparably or worse that SSME (Ours) across the two multiclass tasks and three labeled data settings we consider.

## D.5   Results reporting performance of alternate parameterization

SSME can be fit using other functions to approximate $P(\mathbf{s}|y)$. We additionally explored the value of parameterizing each component using a normalizing flow (implementation details in Sec. E.3), a technique that is increasingly used to approximate complex densities [2].

We find that using a KDE outperforms the normalizing flow on each binary task, for every metric (except estimating AUPRC for SARS-CoV inhibition prediction, Table S15). We find that the KDE performs similarly to the normalizing flow for each LLM-based task, although by a smaller margin (Table S14). We hypothesize that this is due to the established challenges of approximating multi-modal distributions with normalizing flows [65, 16].

For higher-dimensional tasks, the normalizing flow seems to outperform the KDE. In particular, we ran an experiment on ImagenetBG, which contains nine classes and four classifiers. Table S16 demonstrates that in this setting, the normalizing flow performs better than the KDE.

| Dataset | $n_\ell$ | $n_u$ | Labeled | SSME-M | SSME (Ours) | SSME-NF |
|---|---|---|---|---|---|---|
| Critical Outcome | 20 | 1000 | 5.19 ± 1.07 | 1.70 ± 0.27 | **0.67 ± 0.13** | 9.80 ± 1.41 |
| | 50 | 1000 | 2.90 ± 0.59 | 1.65 ± 0.25 | **0.78 ± 0.13** | 8.81 ± 1.11 |
| | 100 | 1000 | 2.09 ± 0.41 | 1.30 ± 0.20 | **0.77 ± 0.13** | 8.28 ± 1.64 |
| ED Revisit | 20 | 1000 | 5.11 ± 0.98 | 1.64 ± 0.34 | **0.45 ± 0.10** | 3.54 ± 1.87 |
| | 50 | 1000 | 2.02 ± 0.58 | 1.46 ± 0.27 | **0.53 ± 0.11** | 2.63 ± 1.10 |
| | 100 | 1000 | 1.43 ± 0.32 | 1.18 ± 0.25 | **0.57 ± 0.11** | 2.51 ± 0.94 |
| Hospital Admission | 20 | 1000 | 7.32 ± 1.25 | 3.29 ± 0.47 | **1.88 ± 0.29** | 3.19 ± 0.95 |
| | 50 | 1000 | 5.40 ± 0.83 | 3.17 ± 0.51 | **1.95 ± 0.27** | 2.27 ± 0.62 |
| | 100 | 1000 | 3.64 ± 0.55 | 3.06 ± 0.45 | **1.51 ± 0.23** | 3.66 ± 2.31 |
| SARS-CoV Inhibition | 20 | 1000 | 6.11 ± 0.96 | 3.06 ± 0.23 | **2.30 ± 0.16** | 11.67 ± 5.26 |
| | 50 | 1000 | 3.22 ± 0.57 | 2.59 ± 0.26 | **2.35 ± 0.10** | 20.89 ± 7.38 |
| | 100 | 1000 | 2.04 ± 0.38 | **1.84 ± 0.24** | 2.36 ± 0.13 | 9.89 ± 4.52 |
| Toxicity Detection | 20 | 1000 | 5.95 ± 0.73 | 6.71 ± 0.23 | **2.34 ± 0.15** | 3.44 ± 0.11 |
| | 50 | 1000 | 4.03 ± 0.68 | 5.38 ± 0.28 | **2.22 ± 0.13** | 3.31 ± 0.13 |
| | 100 | 1000 | 2.43 ± 0.41 | 3.80 ± 0.32 | **2.14 ± 0.15** | 3.32 ± 0.13 |

Table S15: **Mean absolute error in accuracy estimation on binary tasks, including NF parameterization.** We include columns for the Labeled, SSME fit to each classifier individually (SSME-M), SSME parameterized by a KDE (SSME (Ours)), and SSME parameterized by an NF (SSME-NF). While SSME-NF sometimes outperforms SSME-M, SSME-NF never outperforms SSME-KDE on accuracy estimation in any of the tasks or amounts of labeled data we consider.

| Dataset | $n_\ell$ | $n_u$ | Labeled | Majority-Vote | PL | DS | AutoEval | Active-Testing | SSME (KDE) | SSM |
|---------|------|------|---------|---------------|-----|-----|----------|----------------|------------|-----|
| ImnetBG | 20 | 1000 | $6.62 \pm 2.74$ | $2.99 \pm 0.90$ | $33.45 \pm 2.96$ | $5.78 \pm 0.71$ | $6.55 \pm 2.62$ | $10.83 \pm 5.34$ | $8.76 \pm 1.00$ | **2.65** |
| ImnetBG | 50 | 1000 | $3.98 \pm 1.63$ | $3.01 \pm 0.61$ | $17.88 \pm 2.78$ | $5.69 \pm 0.73$ | $3.87 \pm 1.56$ | $12.25 \pm 7.28$ | $8.18 \pm 0.90$ | **2.66** |
| ImnetBG | 100 | 1000 | $2.97 \pm 1.38$ | $2.73 \pm 0.57$ | $9.37 \pm 1.53$ | $5.34 \pm 0.63$ | $2.73 \pm 1.13$ | $9.08 \pm 4.22$ | $8.02 \pm 0.90$ | **2.10** |

Table S16: Comparison across evaluation methods on the *ImnetBG* dataset with varying numbers of labeled examples ($n_\ell$) and unlabeled examples ($n_u$). SSME (NF) consistently achieves the lowest MAE.

| Labeled Examples | Initialization | MAE, ECE | MAE, Accuracy |
|------------------|----------------|----------|---------------|
| 20 | KNN | 0.045 | 0.046 |
| 20 | Draw | 0.035 | 0.034 |
| 50 | KNN | 0.037 | 0.040 |
| 50 | Draw | 0.043 | 0.043 |
| 100 | KNN | 0.031 | 0.033 |
| 100 | Draw | 0.037 | 0.036 |

Table S17: Performance comparison across different numbers of labeled examples and initialization methods (Dataset: CivilComments).

### D.6 Results reporting robustness to initializations

We find that SSME's performance remains strong when using an alternative plausible initialization method: using a majority vote among k=5 nearest neighbors among the labeled examples to initialize cluster assignment for unlabeled examples. We compare this alternate initialization method to our original initialization method (which we refer to as "draw" in the table) on the CivilComments dataset with three different amounts of labeled data (20, 50, and 100 points). We include our results on ECE and accuracy estimation error in the table S17, where we find that the two initialization methods perform comparably on the CivilComments dataset.

### D.7 Results reporting performance as number of classifiers increases

As shown in table S18, we find that SSME's performance improves as the number of classifiers increases, which accords with our theoretical results. As a caveat, our theory predicts that SSME is likely not to perform well if one adds an inaccurate (e.g., worse than random) classifier to the set.

We also find that SSME's performance does not vary greatly across classifier sets, suggesting that SSME's performance is robust to using different classifier sets. In particular, the standard deviation in accuracy estimation error across random classifier sets is always under 0.011, and the standard deviation in ECE estimation error is always under 0.015, indicating consistent performance across classifier sets.

First, we find that performance does not vary greatly across classifier sets, suggesting that SSME's performance is robust to using different classifiers. In particular, the standard deviation in accuracy estimation error across random classifier sets is always under 0.011, and the standard deviation in ECE estimation error is always under 0.015.

Second, we find that SSME's performance improves as the number of classifiers increases, which accords with our theoretical results. As a caveat, our theory predicts that SSME is likely not to perform well if one adds an inaccurate (e.g., worse than random) classifier to the set.

### D.8 Results reporting performance for different bandwith selection procedures

We ran an experiment to test the robustness of our results to the choice of bandwidth. In the paper, we use the Sheather-Jones algorithm to automatically identify a bandwidth. We compare this approach to (1) another automated bandwidth selection algorithm (the Silverman algorithm) and (2) two fixed bandwidths, chosen to be larger but within an order of magnitude of the Sheather-Jones selected bandwidth. We conduct experiments using the CivilComments dataset and 3 labeled dataset sizes

| # Classifiers | MAE, ECE (SD) | MAE, Accuracy (SD) |
|---|---|---|
| 2 | 0.0457 (0.0059) | 0.0471 (0.0047) |
| 3 | 0.0302 (0.0107) | 0.0308 (0.0098) |
| 4 | 0.0268 (0.0094) | 0.0273 (0.0096) |
| 5 | 0.0252 (0.0072) | 0.0253 (0.0078) |
| 6 | 0.0235 (0.0043) | 0.0235 (0.0048) |
| 7 | 0.0225 (only one subset) | 0.0223 (only one subset) |

Table S18: Performance as the number of classifiers increases. Standard deviations (SD) are shown in parentheses. Dataset: CivilComments.

| # Labeled Examples | Bandwidth Selection Procedure | MAE, ECE | MAE, Accuracy |
|---|---|---|---|
| 20 | Sheather–Jones | 0.024 | 0.023 |
| 20 | Silverman | 0.040 | 0.038 |
| 20 | 1.0 | 0.037 | 0.036 |
| 20 | 2.0 | 0.046 | 0.044 |
| 50 | Sheather–Jones | 0.023 | 0.022 |
| 50 | Silverman | 0.066 | 0.065 |
| 50 | 1.0 | 0.062 | 0.062 |
| 50 | 2.0 | 0.045 | 0.044 |
| 100 | Sheather–Jones | 0.022 | 0.021 |
| 100 | Silverman | 0.049 | 0.047 |
| 100 | 1.0 | 0.056 | 0.055 |
| 100 | 2.0 | 0.034 | 0.032 |

Table S19: (Dataset: *CivilComments*) Comparison of bandwidth selection procedures. Sheather–Jones outperforms another automated selection method (Silverman) and two fixed bandwidths.

(20, 50, and 100 labeled datapoints). While results remain strong in all cases, we achieve the best performance when using the Sheather-Jones bandwidth selection procedure.

### D.9 Comparison to baselines drawn from weak supervision

Popular approaches to weak supervision including Snorkel [58] and FlyingSquid [27] implement a latent variable model equivalent to Dawid-Skene. Both works build on Dawid-Skene to incorporate information about pairwise correlations between labeling functions; [58] employs a technique to infer dependencies, while [27] assume these dependencies to be user-provided. When we applied a standard approach to dependency inference [4] in our setting, we observed that (1) all classifiers are inferred to be dependent on one another, and (2) the number of dependencies raised issues with convergence. It is thus not feasible to incorporate dependency inference, and the resulting latent variable model is equivalent to Dawid-Skene.

### D.10 Comparison to ensembling

While we limit the scope of our experiments in the main text to semi-supervised methods that make use of *both* labeled and unlabeled data, another approach would be to produce an estimate of $Pr(y = k|s^{(i)})$ by averaging the classifier scores. This approach results in an unbiased metric estimator when the resulting ensemble is calibrated, as theoretical results by [37] show. Such an approach has natural downsides: it is sensitive to the composition of the classifier set, does not improve with the introduction of labeled data, and relies on an assumption of ensemble calibration that is unlikely to hold in practice [71]. Here, we provide experiments to illustrate this behavior.

We use a semisythetic setting to conduct a comparison to ensembling. To do so, we create sets of three classifiers based on the widely-used Adult dataset [7], where the task is to predict whether a person's income is above $50K. To create differences between the three classifiers in a set, we train them on random fixed-size samples of 50 labeled examples from different portions of the dataset, partitioned based on age. In doing so, our semi-synthetic classifier sets mimic how training data for

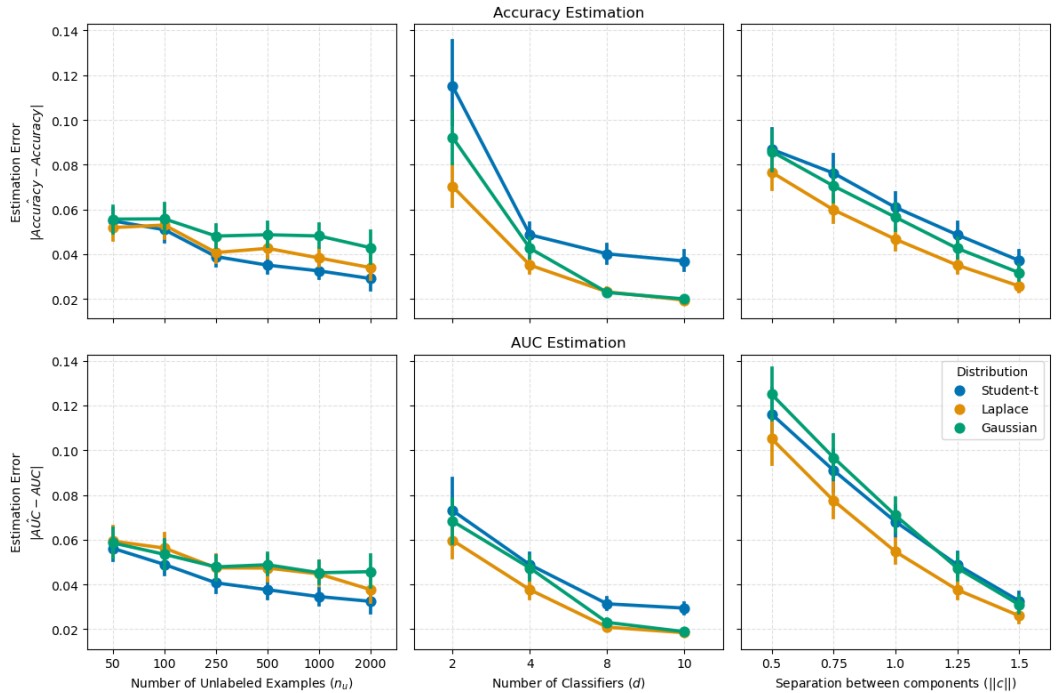

Figure S2: **SSME performance in response to additional unlabeled data, additional classifiers, and improved classifiers.** As indicated by our theoretical results, SSME benefits from increased amounts of unlabeled data, number of classifiers, and classifier performance.

different real-world classifiers can differ in meaningful ways. We repeat this procedure to produce 500 sets of three classifiers, where sets differ in the training data provided to each classifier. Our procedure naturally produces random variation in classifier properties, like accuracy and calibration. As with the real datasets, we produce three splits: a training split to learn the classifiers (50 examples), an estimation split for the performance estimation methods (20 labeled examples and 1000 unlabeled examples), and an evaluation split to measure ground truth values for each metric (10,000 examples). Each classifier is a logistic regression with default L2 regularization.

We artificially increase the expected calibration error of each classifier using a generalized logistic function parameterized by $a$. Specifically, we transform classifier score $s$ to be $\frac{s^a}{s^a+(1-s)^a}$, effectively increasing overconfidence for higher $s$ and increasing underconfidence for lower $s$. As in the semisynthetic experiments, we generate 500 semisynthetic classifier sets, where each classifier in a set is trained on 100 examples distinct from the training data for other classifiers in the set (results are robust to this choice of training dataset size). Each set contains three classifiers.

Figure S3 reports our results. As the average calibration among classifiers in a set varies, SSME consistently improves over the use of an ensemble. This aligns with our intuition, and indicates the value of using labeled data in conjunction with unlabeled data. Miscalibration has little effect on the ensemble when estimating AUPRC; here, SSME and ensembling perform similarly.

### D.11   Comparison of computational cost

Fitting SSME is computationally cheap and faster than the best baseline. For every dataset and experiment configuration reported in the paper, SSME can be fit in under 5 minutes, using only 1 CPU. SSME inherits the computational complexity of kernel density estimators ($O(n^2)$) and EM. In a matched comparison (using 20 labeled examples, 1000 unlabeled examples, and 7 classifiers) the next best-performing baseline (Bayesian-Calibration) takes on average 32.1 seconds to estimate performance, while SSME takes 21.5 seconds.

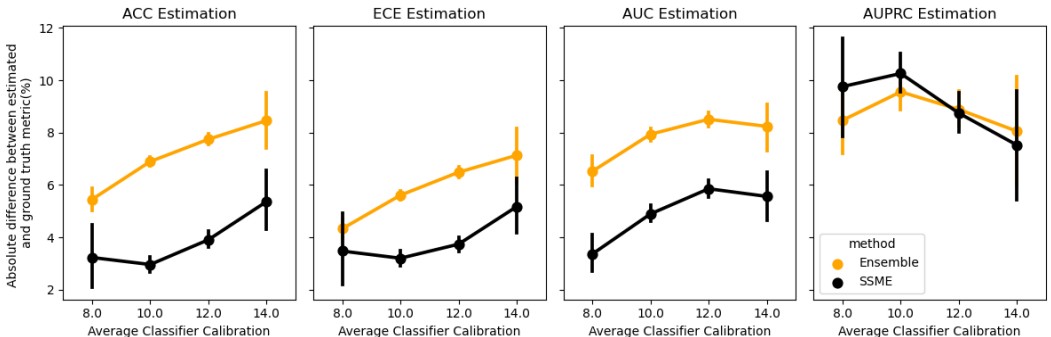

Figure S3: **A comparison of SSME to ensembling on a miscalibrated classifier set.** SSME consistently produces more accurate performance estimates compared to ensembling the classifiers across differently calibrated classifier sets (x-axis).

---

**Algorithm 2:** Sampling-based metric estimation with SSME

---

**Input:** Labeled set $\mathcal{D}_\ell = \{(\mathbf{s}^{(i)}, y^{(i)})\}_{i=1}^{n_\ell}$; Unlabeled set $\mathcal{D}_u = \{\mathbf{s}^{(i)}\}_{i=1}^{n_u}$; $\mathbf{s}^{(i)}$ are **ALR-transformed** classifier scores; Fitted model $P_\theta(y \mid \mathbf{s})$; metric function $g(\mathbf{p}, y)$; Number of samples $S = 500$

**Init:** sum $\leftarrow 0$;

**1 for** $s \leftarrow 1$ **to** $S$ **do**

**2**     **foreach** $\mathbf{s}^{(i)} \in \mathcal{D}_u$ **do**

**3**         $\tilde{y}^{(i)} \sim P_\theta(y \mid \mathbf{s}^{(i)});$;                    `// Sample true label` $y$ `from mixture model`

**4**     $\mathcal{D}^{(s)} \leftarrow \{(\mathbf{s}^{(i)}, \tilde{y}^{(i)})\} \cup \mathcal{D}_\ell$;

**5**     $\texttt{metric}^{(s)} \leftarrow (n_u + n_\ell)^{-1} \sum_{(\mathbf{s}, y) \in \mathcal{D}^{(s)}} g(\mathrm{ALR}^{-1}(\mathbf{s}), y)$;

**6**     $\texttt{sum} \leftarrow \texttt{sum} + \texttt{metric}^{(s)}$;

**Output:** $\widehat{\texttt{metric}} = \texttt{sum}/S$ ;                    `// Sample-based estimate of metric`

---

## E   Method Details

### E.1   Additive Log-Ratio Transform

Given a vector $p \in \Delta^{K-1}$ over $K$ classes, let $\mathbf{s} = \mathrm{ALR}(\mathbf{p}) = \left[\log \frac{\mathbf{p}_1}{\mathbf{p}_K}, \log \frac{\mathbf{p}_2}{\mathbf{p}_K}, \cdots, \log \frac{\mathbf{p}_{K-1}}{\mathbf{p}_K}\right] \in \mathbb{R}^{K-1}$. To invert, $\mathbf{p}_i = \frac{e^{\mathbf{s}_i}}{1 + \sum_{k=1}^{K-1} e^{\mathbf{s}_k}}$ for $i < K$ and $\mathbf{p}_K = \frac{1}{1 + \sum_{k=1}^{K-1} e^{\mathbf{s}_k}}$. The ALR transform maps unit-sum data into real space, where it is easier to fit mixture models. The inverse allows us to map samples from the mixture model in real space back to the simplex $\Delta^{K-1}$. For details, see [53].

### E.2   Metric Estimation

SSME is able to estimate any metric that is a function of the classifier probabilities $p$ and label $y$. We approximate the joint distribution $P(y, \mathbf{p})$ with a mixture model model $P_\theta(y, \mathbf{s})$, where $\mathbf{s}$ refers to the ALR-transformed classifier probabilities (i.e. "classifier scores")[2]. We refer to $P(y, \mathbf{p})$ for ease of notation in this section; it is equivalent, through invertible mapping, to $P(y, \mathbf{s})$.

We denote our approximation for $P(\mathbf{p}, y)$ as $P_\theta(\mathbf{p}, y)$. We provide a few concrete examples of how one can use SSME to measure performance metrics, given $P_\theta(\mathbf{p}, y)$ and a set of unlabeled probabilistic predictions $\{\mathbf{p}^{(i)}\}_{i=1}^{n_u}$ and labeled probabilistic predictions $\{\mathbf{p}^i, y^{(i)}\}_{i=1}^{n_\ell}$. Notationally, $\mathbf{p}_j^i$ refers to the $j$th model's probabilistic prediction of the $i$th unlabeled example.

---

[2]Recall that ALR is a bijection, so we use the inverse mapping $\mathrm{ALR}^{-1} : \mathbb{R}^{K-1} \to \Delta^{K-1}$ to transform our mixture distribution in real space back to probability space.

**Accuracy** measures the alignment between a model's (discrete) predictions and the true label $y$. To discretize predictions, practitioners typically take the argmax of $\mathbf{p}^{(i)}$. Using the binary case an illustrative example, the accuracy of the $j$th model can be written as:

$$\text{Accuracy}_j = \mathbb{E}_{\mathbf{p}}\left[\mathbf{1}\left[y = \mathbf{1}(\mathbf{p} > t)\right]\right]$$

where $\mathbf{1}$ is an indicator function and $t$ is a chosen threshold, typically 0.5. In our setting, we approximate this as:

$$\text{Accuracy}_j \approx \frac{1}{n_u + n_\ell} \sum_{i=1}^{n_u + n_\ell} \mathbf{1}\left[y^{(i)} = \mathbf{1}(\mathbf{p}^{(i)} > t)\right]$$

For labeled examples, we use the true label $y^{(i)}$. For unlabeled examples, we draw $y^{(i)} \sim P_\theta(y|\mathbf{p}^{(i)})$. We then compute accuracy using these labels $y^{(i)}$ and predictions $\mathbf{p}^{(i)}$. To ensure our estimation procedure is robust to sampling noise, we average our estimated accuracy over 500 separate sampled labels for each example in the unlabeled dataset.

Alternatively, we could directly use $P_\theta(y|\mathbf{p})$ to estimate accuracy. That is, for each point $\mathbf{p}^{(i)}$ we directly compute an expectation for the label, and sum this over the entire dataset.

Using the binary case as an example

$$\text{Accuracy}_j \approx \frac{1}{n_u + n_\ell} \sum_{i=1}^{n_u + n_\ell} \mathbb{E}\left[\mathbf{1}\left[y^{(i)} = \mathbf{1}(\mathbf{p}_j^{(i)} > t)\right] | \mathbf{p}^{(i)}\right]$$

In other words, we compute the expectation that the true label agrees with the predicted label for each point . This expectation is $\mathbf{p}^{(i)}$. This expectation is computed over $P_\theta(y|\mathbf{p})$ One can interpret $P_\theta(y|\mathbf{p})$ as a "recalibration" step: given a set of classifier guesses $\mathbf{p}$, what is the true distribution of $y$?

In our experiments, we use the first of these two approaches, i.e. we sample the true label from the estimated distribution.

**Expected Calibration Error (ECE)** measures the alignment between a model's predicted probabilities $\mathbf{p}_j$ and the ground truth labels $y$. In particular, ECE compares the model's reported confidence to the true class likelihoods, averaged over the dataset. We write out our ECE estimation procedure for the binary case, and it extends readily to definitions of calibration in multiclass settings [32]. Binary ECE can be written as:

$$\text{ECE}_j = \mathbb{E}_{\mathbf{p}_j}\left[\left|P(\hat{Y} = 1|\hat{p} = \mathbf{p}_j) - \mathbf{p}_j\right|\right]$$

Then, to approximate the ECE with the datasets $\{\mathbf{p}^i\}_{i=1}^{n_u}$ and $\{\mathbf{p}^i, y^{(i)}\}_{i=1}^{n_\ell}$, one can sample $y^{(i)} \sim P_\theta(y|\mathbf{p}^{(i)})$ for each unlabeled sample $i$ and then use the standard histogram binning procedure [31] using both the true labels for the labeled dataset and the sampled labels for the unlabeled dataset. In this approach, we treat the sampled labels $y^{(i)}$ as true labels for unlabeled examples. To ensure our procedure is robust against sampling noise, we draw samples of $y^{(i)}$ repeatedly for a fixed number of draws (500). We then compute ECE separately for each of these 500 draws and average ECE across all draws.

We use this first approach, but alternatively, one could also *directly* use $P_\theta(y|\mathbf{p})$ to estimate ECE. In particular, we can write:

$$\text{ECE}_j \approx \frac{1}{n_u + n_\ell} \sum_{i=1}^{n_u + n_\ell} \left|P_\theta\left(y = 1|\mathbf{p}_j^{(i)}\right) - \mathbf{p}_j^{(i)}\right|$$

In this approach, we don't sample the labels $y$ for unlabeled examples but instead directly use $P_\theta(y|\mathbf{p})$, which provides us (an estimate of) the true distribution of $y$. Instead, we directly use our estimate for the conditional label distribution $P_\theta\left(y = 1|\mathbf{p}_j^{(i)}\right)$.

In our experiments, we use the first approach described, i.e. sampling $y^{(i)}$ for unlabeled examples and then using the standard binning and averaging procedure.

**AUROC and AUPRC** can be estimated with a similar procedure as above. In particular, we sample a label $y^{(i)} \sim P_\theta \left( y = 1 | \mathbf{p}^{(i)} \right)$ from the conditional label distribution and compare these sampled labels to the classifier probabilities.

### E.3 Alternate parameterizations

One alternative parameterization is to use a normalizing flow to model our mixture of distributions. Normalizing flows learn and apply an invertible transform $f_\theta$ to a random variable $\mathbf{z} \sim D_1$ to obtain $f_\theta(\mathbf{z}) \sim D_2$. Here, we set $\mathbf{z} \sim D_1$ to a Gaussian mixture model and learn a transformation such that $f_\theta(\mathbf{z}) \overset{\text{dist.}}{\approx} \mathbf{s}$, i.e., the transformed distribution roughly matches our classifier score distribution. By modeling $\mathbf{z}$ explicitly as a Gaussian mixture model, one can move back and forth between the two distributions, as $f_\theta^{-1}(f_\theta(\mathbf{z})) = \mathbf{z}$, where $f_\theta^{-1}$ is the inverse of $f$. Specifically, we set the distribution of $\mathbf{Z}$ to follow a Gaussian mixture:

$$\mathbf{Z}|(Y = k) \sim \mathcal{N}(\mu_k, \Sigma_k)$$

Thus, the marginal distribution of $\mathbf{Z}$ is $p_\mathbf{Z}(\mathbf{z}) = \sum_{k=1}^{K} \mathcal{N}(\mathbf{z}|\mu_k, \Sigma_k) \cdot p(y = k)$ is the overall density of $\mathbf{z}$. We apply our invertible transformation $f_\theta$ to obtain $\mathbf{s} = f_\theta(\mathbf{z})$. To find $p(\mathbf{s}|y = k)$, we follow the approach of [35]:

$$p_\mathbf{S}(\mathbf{s}|y = k) = \mathcal{N}(f_\theta^{-1}(\mathbf{s})|\mu_k, \Sigma_k) \cdot \left| \det\left( \frac{\delta f}{\delta x} \right) \right| \cdot p(y = k)$$

Intuitively, we transform $(\mathbf{s}, y)$ into a distribution $(\mathbf{z}, y)$ which follows a Gaussian mixture model. By enforcing the constraint that this transform is invertible, the joint distribution on $(\mathbf{z}, y)$ captures all the information in $(\mathbf{s}, y)$.

We use the RealNVP architecture [22] to parameterize $f_\theta$ using 10 coupling layers, 3 fully-connected layers, and a hidden dimension of 128 between the fully connected layers. Our normalizing flow is lightweight and trains in less than a minute for each dataset in our experiments section using 1 80GB NVIDIA A100 GPU.

Note there are two optimizations here: (1) the normalizing flow transformation $f_\theta$ which maps $\mathbf{s}$ into our latent Gaussian mixture space and (2) the Gaussian mixture model parameters $\mu_k, \Sigma_k$ themselves. We begin by fixing the GMM parameters $\mu_k, \Sigma_k$ to values estimated from our classifier scores $\mathbf{s}$ and learning only the flow $f_\theta$ for 300 epochs. Afterwards, we optimize the GMM parameters $\mu_k, \Sigma_k$ with EM for another 700 epochs.

