# OpenReview forum: "Evaluating multiple models using labeled and unlabeled data"
_NeurIPS.cc/2025/Conference — NeurIPS 2025 poster_

### Official Review · Reviewer_4KCW · 2025-06-29

**Clarity:** 3
**Significance:** 4
**Originality:** 4
**Rating:** 4
**Confidence:** 4

**Summary:**

This paper introduces Semi-Supervised Model Evaluation (SSME), a method that leverages both labelled and unlabeled data to improve evaluation accuracy. SSME uses a semi-supervised mixture model to estimate the joint distribution of ground truth labels and classifier scores, reducing error through the combination of multiple classifiers and probabilistic scores. Authors perform experiments across diverse domains to demonstrate that SSME provides more accurate estimates than existing methods, reducing error by 5.1x relative to labelled data and 2.4x relative to the next best method.

**Questions:**

How does the author’s contribution improve the state of the art?
How do authors ensure the lack of data leaks in the datasets they used?

**Ethical Concerns:**

["NO or VERY MINOR ethics concerns only"]

**Final Justification:**

I discussed the paper with the authors. However, while some issues were resolved, there are some limitations of the work that the authors won't be able to further address here. These limitations include poor scalability to high-dimensional data. This would be important to tackle in this day and age. Given my generally high initial score and in the face of hard limitations of the proposed methodology, I will retain my fairly high score for this work.

**Limitations:**

Limitations are poorly discussed in the manuscript. I suggest outlining a specific section.

**Paper Formatting Concerns:**

It would be better to have related work introduced right after the problem setting, not at the end of the manuscript. This needs to be addressed before acceptance.

Appendix is missing bibliography. If those are supposed to be the references from the main manuscript - the are miscited as they don't correspond

**Quality:**

2

**Strengths And Weaknesses:**

Strength:
The authors propose an approach for an extremely poignant problem in ML and Data Science - the lack of well-annotated datasets. Even though new datasets appear constantly, there's arguably never enough data. Therefore searching for alternative approaches is important. The manuscript seems to be sound theoretically.

Weaknesses:

1) By their own admission in the checklist, the concept of the authors does not outperform the benchmarks. The advantage of the contribution approach has to be made clear.

2) Overall, the concept seems to borrow from the fields of semi-supervised learning and active learning. However, the existing concepts have not been sufficiently discussed in the related work section.

3) The writing of the manuscript and the appendix could be significantly improved: it is impossible to verify the sources and adequacy of the tasks.

4) Furthermore, the supplement does not include the source code, despite the statement in the checklist. Lack of source code and clear references to the datasets limits my ability to review this work.

---

> ### Author Rebuttal · Authors · 2025-07-31
>
> Thank you for your comments and the positive review! We appreciate that you recognize the importance of semi-supervised evaluation and believe our method is theoretically sound.
>
> We address your questions and respond to the highlighted weaknesses below.
>
> **Q1. How does the author’s contribution improve the state of the art? How do authors ensure the lack of data leaks in the datasets they used?**
>
> Our approach improves upon the state-of-the-art in semi-supervised performance estimation by framing the problem as semi-supervised mixture model estimation; we show empirically that our method outperforms eight baselines across a wide range of tasks and metrics. We ensure there is no dataset leakage by using distinct sets of data to train the classifiers (the training split), to estimate performance (the estimation split), and to measure ground truth performance (the evaluation split). We detail this procedure in Section B.1 and will better incorporate this information into the main text.
>
> **Q2. Adding a limitations section**
>
> Thank you for this suggestion. We will add a dedicated limitations section to clearly discuss the scope and potential constraints of our approach, including the difficulty of estimating KDEs in high dimensions and the assumption that the labeled and unlabeled data are drawn from the same distribution.
>
> **Q3. Paper organization: moving the related work to be right after the problem setting. Addressing missing bibliography for the appendix.**
>
> Thank you for the suggestion and for flagging the reference issue in the appendix. We will shift the related work to appear after the introduction. The appendix references map to the main text bibliography (we manually split the files to submit the supplementary material). We will ensure that the references properly accompany the appendix.
>
>
> **W1. Outperforming baselines**
>
> We’d like to clarify: SSME consistently outperforms the eight baselines we compare to across metrics. In particular, SSME outperforms the next-best baseline in every metric tested (2.1x gain in accuracy estimation, 1.5x gain in ECE estimation, 1.2x gain in AUC estimation, and 1.1x gain in AUPRC estimation) averaged across tasks. We will clarify our statement in the checklist, which was intended to transparently state that we discuss limitations of our work and settings where baselines could outperform SSME (e.g. at Line 310 we discuss settings that would favor Bayesian-Calibration). Overall, however, we decisively outperform all baselines we compare to.
>
> **W2. Clarifying relationship to work in semi-supervised learning and active learning**
>
> Thanks for raising this! Our work indeed builds off prior work in semi-supervised learning and, to a lesser extent, active learning. All the baselines we compare to use either semi-supervised (Bayesian-Calibration, AutoEval, SPE, Dawid-Skene) or active learning (Active-Testing) to derive their performance estimates. Unlike prior methods, we are able to simultaneously incorporate continuous scores, labeled and unlabeled data, and multiple classifiers, whereas previous works only use at most two of these three data sources. We will further clarify this point in the related work.
>
>
> **W3. Improve writing to make clear the sources of the tasks; include references to datasets.**
>
> Thanks for the suggestion; while we do include references for each dataset in Section 4.1, we will make the connections more explicit. Our code, once public, will also support downloading and processing each of the five public datasets used in our experiments.
>
> **W4. Inclusion of source code**
>
> Our apologies, we misunderstood the checklist to mean the code must be made public upon publication. We will make the code available upon publication, including all code required to reproduce experimental results on five public datasets. Unfortunately, NeurIPS reviewing guidelines do not allow anonymized code URLs during the rebuttal process, so we cannot provide the code right now. The codebase will also support practitioners who wish to apply SSME to new settings.
>
> **Please let us know if our response addresses your primary concerns; if so, we would kindly appreciate it if you could raise your score. If not, are there additional concerns we can address?**

---

> > ### Comment · Reviewer_4KCW · 2025-08-05
> >
> > I thank the authors for the discussion. It seems however, that there are some limitations of the work that the authors won't be able to further address here. These limitations include poor scalability to high-dimensional data. This would be important to tackle in this day and age. Given my generally high initial score and in the face of hard limitations of the proposed methodology, I will retain my fairly high score for this work.

---

> > > ### Author Response · Authors · 2025-08-06
> > > **Author Response**
> > >
> > > Thank you for engaging with our rebuttal, and for your helpful feedback. To clarify, SSME is a general framework whose core proposal is to estimate the joint distribution of labels and model predictions, P(y, s). While our main experiments use KDEs to estimate P(y, s) (which perform well empirically), this choice is not a hard limitation. SSME can accommodate any other technique to estimate P(y, s), including scalable alternatives like normalizing flows. We evaluate such a parametrization in Appendix D5.
> > >
> > > In response to your concern, we conducted a new experiment exploring SSME’s performance in a higher-dimensional setting (ImageNetBG, containing 9 classes, with four classifiers). The table reports the MAE in accuracy estimation (averaging across classifiers) with standard deviations computed over 50 random samples of performance estimation data.  We found that SSME parametrized by a normalizing flow outperforms all baselines  (see table below) and also outperforms SSME parametrized by a KDE. These results suggest that normalizing flows provide a promising alternative to KDEs in high-dimensional semi-supervised performance estimation problems. We will include these results in the paper along with a discussion of their implications.
> > >
> > > **If these additional experiments address your concerns, we would appreciate it if you’d consider updating your score.**
> > >
> > > | Dataset     | n_ℓ | n_u  | Labeled         | Majority-Vote   | PL | DS     | AutoEval         | Active-Testing   | SSME (KDE)        | SSME (NF)        |
> > > |-------------|-----|------|------------------|------------------|------------------|------------------|------------------|------------------|------------------|-----------------|
> > > | ImnetBG  | 20  | 1000 | 6.62 ± 2.74      | 2.99 ± 0.90      | 33.45 ± 2.96     | 5.78 ± 0.71      | 6.55 ± 2.62      | 10.83 ± 5.34     | 8.76 ± 1.00      | 2.65 ± 0.67     |
> > > | ImnetBG  | 50  | 1000 | 3.98 ± 1.63      | 3.01 ± 0.61      | 17.88 ± 2.78     | 5.69 ± 0.73      | 3.87 ± 1.56      | 12.25 ± 7.28     | 8.18 ± 0.90      | 2.66 ± 0.81     |
> > > | ImnetBG  | 100 | 1000 | 2.97 ± 1.38      | 2.73 ± 0.57      | 9.37 ± 1.53      | 5.34 ± 0.63      | 2.73 ± 1.13      | 9.08 ± 4.22      | 8.02 ± 0.90      | 2.10 ± 0.68     |

---

> > > > ### Comment · Reviewer_4KCW · 2025-08-07
> > > >
> > > > Thank you for the discussion and additional results.

---

### Official Review · Reviewer_HJAd · 2025-06-30

**Clarity:** 3
**Significance:** 2
**Originality:** 3
**Rating:** 4
**Confidence:** 4

**Summary:**

The paper introduces Semi-Supervised Model Evaluation (SSME), a simple EM-based mixture model that learns the joint distribution of true labels and the score vectors of multiple classifiers from a few labeled examples plus abundant unlabeled data; once fitted, this density lets can be used to infer any performance metric (accuracy, AUC, calibration, etc.) on the unlabeled pool. The authors prove that the estimation error shrinks with more unlabeled samples and more diverse classifiers, then show across five real-world domains (including LLM-based text classifiers) that SSME cuts metric-estimation error by about five-fold compared with using labels alone.

**Questions:**

1. It seems that currently, this method is mainly applied for Natural Language classification tasks, how will it performs on image classification tasks?

2. How will the diversity and the coverage of the classifiers being tested impact the estimation accuracy? Intuitively, more classifiers you considered, the more unbias your joint distribution $P(Y,s)$ should be, I encourage authors to conduct a experiments to test this.

**Ethical Concerns:**

["NO or VERY MINOR ethics concerns only"]

**Final Justification:**

My major concerns before rebuttal is the fact that recent and relevant baselines are not compared or discussed, the authors provided somewhat convincing arguments in their rebuttal - their focus in performance estimation and focus beyond model ranking, some of the baselines I mentioned can only achieve the latter. Therefore, I believe there is no significant reason for me to oppose the acceptance of this paper.

**Limitations:**

N.A.

**Paper Formatting Concerns:**

N.A.

**Quality:**

2

**Strengths And Weaknesses:**

**Strengths:**

1. This paper considers a vital yet relatively overlooked problem - model selection under weak supervision.

2. The proposed method seems new and interesting - we are leveraging the fact that we have multiple models to evaluate, and we are using their predictions to predict pseudo labels for evaluation purposes.

3. The experiments contain a diverse setting spanning from logistic regression to language models.

**Weaknesses:**

1. The baselines being considered in this paper are incomplete and outdated, and many relevant works in the domain of unsupervised accuracy estimation are not compared (while some are discussed), to name some examples: [1, 2, 3, 4]. The topic being proposed here has stronger assumptions than unsupervised accuracy estimation; if not compared empirically, then this makes the technique proposed in this paper not very convincing.

2. This method only works for classification tasks, which, to an extent, limits the significance and contribution of the proposed method.

3. The proposed method is consistently worse than some baselines, such as Bayesian Calibration.

[1] Confidence and Dispersity Speak: Characterising Prediction Matrix for Unsupervised Accuracy Estimation, ICML 2023.

[2] Unsupervised Accuracy Estimation of Deep Visual Models using Domain-Adaptive Adversarial Perturbation without Source Samples, ICCV 2023.

[3] Tune it the right way: Unsupervised validation of domain adaptation via soft neighborhood density, ICCV 2021.

[4] Test accuracy vs. generalization gap: Model selection in nlp without accessing training or testing data, KDD 2023.

---

> ### Author Rebuttal · Authors · 2025-07-31
>
> Thank you for your comments! We appreciate that you recognize that SSME is able to utilize multiple sources of information simultaneously, has wide applicability, and outperforms baselines in estimating performance.
>
> We address your questions and respond to the highlighted weaknesses below.
>
> **Q1. It seems that currently, this method is mainly applied for Natural Language classification tasks, how will it performs on image classification tasks?**
>
> Thanks for this question! SSME outperforms baselines across three settings we test: natural language data, tabular data, and graph data. In response to your question, we have added an experiment where we analyze SSME’s performance on image data in the context of CIFAR-3 [1, 2, 3] (i.e., a dataset with three classes from CIFAR-10), where classifiers are models pre-trained on ImageNet and finetuned on CIFAR. We found SSME outperforms baselines that can be applied in this setting, achieving an mean absolute error in accuracy estimation of 4.50 (SD: 2.12), while the next best baseline (labeled data alone) achieves a mean absolute error of 5.33 (SD: 2.77). We find similar results for ECE estimation, where SSME achieves a mean absolute error in ECE estimation of 3.75 (SD: 1.21), while the next best baseline (pseudo-labeling) achieves a mean absolute error of 6.36 (SD: 4.78).
>
> [1] https://ewsn2022.pro2future.at/paper/sessions/ewsn2022-final3.pdf
>
> [2] https://kclpure.kcl.ac.uk/ws/portalfiles/portal/332763034/2501.06066v3.pdf
>
> [3] https://proceedings.mlr.press/v189/pandey23a/pandey23a.pdf
>
> **Q2. How will the diversity and the coverage of the classifiers being tested impact the estimation accuracy? Intuitively, more classifiers you considered, the more unbias your joint distribution  should be, I encourage authors to conduct a experiments to test this.**
>
> Thanks for this question. We ran an experiment to measure the benefit of each additional classifier, as well as how much performance varies across classifier sets, and include the results below. In particular, for each number of classifiers between 2 and 7 (inclusive), we use SSME to estimate performance for each unique set of classifiers (so for 7 classifiers, there is exactly one set; for 4 classifiers, there are 7 choose 4 = 35 sets). We performed this experiment on the CivilComments dataset.
>
> Your intuition is correct: additional classifiers do produce better estimates of performance, with diminishing returns. As shown in the table below, we find that SSME’s performance improves as the number of classifiers increases, which accords with our theoretical results. As a caveat, our theory predicts that SSME is likely not to perform well if one adds an inaccurate (e.g., worse than random) classifier to the set.
>
> We also find that SSME's performance does not vary greatly across classifier sets, suggesting that SSME’s performance is robust to using different classifier sets. In particular, the standard deviation in accuracy estimation error across random classifier sets is always under 0.011, and the standard deviation in ECE estimation error is always under 0.015, indicating consistent performance across classifier sets.
>
> | Number of classifiers |            MAE, ECE (SD) |       MAE, Accuracy (SD) |
> | --------------------: | -----------------------: | -----------------------: |
> |                     2 |          0.0457 (0.0059) |          0.0471 (0.0047) |
> |                     3 |          0.0302 (0.0107) |          0.0308 (0.0098) |
> |                     4 |          0.0268 (0.0094) |          0.0273 (0.0096) |
> |                     5 |          0.0252 (0.0072) |          0.0253 (0.0078) |
> |                     6 |          0.0235 (0.0043) |          0.0235 (0.0048) |
> |                     7 | 0.0225 (only one subset) | 0.0223 (only one subset) |
>
> (Dataset: CivilComments) MAE for ECE and accuracy across classifier set choice, along with the standard deviation in MAE across classifier sets.
>
> **W1. Comparison to additional methods**
>
> Thanks for your comment! In the original manuscript, we compare to and outperform eight state-of-the-art baselines designed for semi-supervised evaluation. We agree that unsupervised accuracy estimation methods are related, and Appendix A describes connections to past work in this literature. However, these baselines cannot be directly applied to estimate the metrics we wish to estimate. For example: three of the baselines you mention ([1], [3], and [4]) discuss heuristics that correlate well with model accuracy, but it is not clear how to map these heuristics to absolute estimates of other metrics like ECE, AUC, and AUPRC, as SSME does. It is also unclear how to extend the ideas in the other baseline you mention [2], which fits a custom pseudo-labeling function to image data specifically, to our setting, where it is unclear what such a pseudo-labeling function should be for a generic dataset. We agree the connection is interesting and believe it merits future work, and will include the citations you provided in Appendix A; thank you for bringing them up!
>
> [1] Confidence and Dispersity Speak: Characterising Prediction Matrix for Unsupervised Accuracy Estimation, ICML 2023.
>
> [2] Unsupervised Accuracy Estimation of Deep Visual Models using Domain-Adaptive Adversarial Perturbation without Source Samples, ICCV 2023.
>
> [3] Tune it the right way: Unsupervised validation of domain adaptation via soft neighborhood density, ICCV 2021.
>
> [4] Test accuracy vs. generalization gap: Model selection in nlp without accessing training or testing data, KDD 2023.
>
> **W2. Limitation to classification**
> While our method primarily focuses on semi-supervised performance estimation in classification, we note that we can very naturally extend our approach to incorporate regression settings as well. The key idea behind our method is to estimate a joint distribution for P(s, y) with a semi-supervised model. In classification settings, it is natural to model this using a mixture model. In a regression case with a continuous y, however, there are other joint density estimators for P(s,y) that could be reasonable, including KDEs or autoencoders. Thanks for flagging this, which we agree is a natural extension of our method and an interesting direction for future work!
>
> **W3. Proposed method is consistently worse than Bayesian-Calibration**
> Thank you for this comment. To clarify, our results show that SSME decisively outperforms Bayesian-Calibration across the diverse datasets that we study. In particular, our results show that SSME outperforms Bayesian-Calibration in every metric tested (2.1x gain in accuracy estimation, 1.5x gain in ECE estimation, 1.2x gain in AUC estimation, and 1.1x gain in AUPRC estimation), averaging across datasets and tasks. Our tables in the appendix provide further details substantiating this result. Bayesian-Calibration also does not generalize to settings with more than two classes, an important limitation SSME overcomes.
>
> **Please let us know if our response addresses your primary concerns; if so, we would kindly appreciate it if you could raise your score. If not, are there additional concerns we can address?**

---

> > ### Comment · Reviewer_HJAd · 2025-08-05
> >
> > Thank the authors for their comprehensive and quality rebuttal, I am generally convinced - however I do believe not comparing with unsupervised accuracy estimation approach might need further justifications in the paper.
> >
> > In addition, I found two more relevant papers [1,2] discusses model selection for SSL that might require comparsion or discussion in the paper.
> >
> > [1] Random matrix analysis to balance between supervised and unsupervised learning under the low density separation assumption, ICML 2023.
> >
> > [2] Towards Realistic Model Selection for Semi-supervised Learning, ICML 2024.

---

> ### Author Response · Authors · 2025-08-06
> **Author Response**
>
> Thank you for your thoughtful engagement with our rebuttal and for the helpful references. We will incorporate these citations into the paper. We cannot compare to these methods on the tasks we consider because neither method directly estimates model performance (though they are applicable to the related and easier task of model selection). We will clarify the distinction between the two tasks in the paper and expand our justification for the baseline comparisons; thanks again for your reply! If you are convinced by our rebuttal, we would kindly appreciate if you could raise your score.

---

### Official Review · Reviewer_Nk8N · 2025-07-02

**Clarity:** 3
**Significance:** 2
**Originality:** 3
**Rating:** 4
**Confidence:** 4

**Summary:**

### Summary

This paper proposes a method called Semi-Supervised Model Evaluation (SSME) for evaluating the performance of multiple machine learning classifiers in scenarios where large amounts of labeled data are not available.

SSME utilizes a small amount of labeled data, a large amount of unlabeled data, and the probabilistic prediction scores of multiple classifiers to estimate performance metrics such as accuracy, AUC, and ECE. By estimating the joint distribution
𝑃
(
𝑦
,
𝑠
)
 of true labels and classifier scores using a semi-supervised mixture model, it enables performance evaluation even for the unlabeled data under the constraint of limited labeled instances.

### Contributions

Proposal of SSME:
Introduces an evaluation framework that jointly leverages labeled data, unlabeled data, and continuous prediction scores from multiple classifiers. It estimates the joint distribution
𝑃
(
𝑦
,
𝑠
)
 using a mixture model, enabling the calculation of various performance metrics.

Theoretical guarantees:
Provides theoretical upper bounds on the estimation error and shows that the error decreases as the number of classifiers or the amount of unlabeled data increases.

Empirical validation:
Demonstrates effectiveness across diverse domains such as healthcare, social media moderation, molecular property prediction, and natural language processing. Compared to using labeled data only, SSME reduces estimation error by up to 5.1 times on average, and by 2.4 times compared to the second-best method.

Practical applicability:
Applies the method to real-world scenarios including classifiers based on large language models and subgroup-level performance evaluation (e.g., fairness), confirming its utility.

**Questions:**

The ideas behind SSME and the proposed methodology in this paper are intriguing. To better understand the contributions of this work, I would like to ask the following questions:

The paper presents experiments across five tasks, with classifier sets selected from existing benchmarks. However, I believe the choice of classifiers can significantly influence the performance of the proposed method. Do the authors consider the selection of classifiers to be a practical issue that is adequately addressed in the paper, or do they believe it is straightforward enough not to pose a concern?

As the authors themselves note, the proposed method assumes that labeled and unlabeled data are drawn from the same distribution. In practice, this assumption may often be violated. Do the authors believe that such distributional mismatch would not pose a serious issue in real-world applications? Alternatively, do they see this as a limitation that could be addressed through future methodological improvements?

**Ethical Concerns:**

["NO or VERY MINOR ethics concerns only"]

**Final Justification:**

After reviewing the authors' rebuttal, the questions and concerns I had as a reviewer have been resolved, so I will improve my evaluation by one rank.

**Limitations:**

yes

**Paper Formatting Concerns:**

No major formatting issues were observed.

**Quality:**

3

**Strengths And Weaknesses:**

**Strengths**

* **Utilization of Rich Data Sources**: SSME leverages three sources of information simultaneously—limited labeled data, abundant unlabeled data, and multiple machine learning classifiers—which conventional evaluation methods struggle to use effectively. This integrated approach significantly improves estimation accuracy in classifier evaluation.

* **High Accuracy in Performance Estimation**: SSME estimates classifier performance more precisely than competing methods. For example, it reduces estimation error by a factor of 5.1 compared to using labeled data alone, and by 2.4 times compared to the second-best method. Notably, in estimating Expected Calibration Error (ECE), it achieves an average error reduction of 7.2 times.

* **Wide Applicability and Versatility**: SSME supports the estimation of various performance metrics such as Accuracy, AUC, AUPRC, and ECE. It is applicable to any number of classifiers and classes. The method has demonstrated its utility across diverse domains including healthcare, content moderation, molecular property prediction, and text classification. It also performs well in evaluating classifiers derived from large language models (LLMs) and in subgroup-specific performance evaluations.

**Weaknesses**

* **Simplifying Assumptions in Theoretical Analysis**: The theoretical results in the paper are based on stylized settings, such as assuming classifier scores are generated from Gaussian mixture models. While empirical evidence supports the trends predicted by the theory, its direct applicability to real-world scenarios may be limited if actual data deviates significantly from these assumptions.

* **Assumption of Identical Data Distributions**: The experiments assume that unlabeled data are sampled from the same distribution as labeled data. However, in real-world settings, labeled and unlabeled data may systematically differ (e.g., covariate shift), which may limit the direct applicability of the current version of SSME.

* **Performance Variability in Specific Metrics or Scenarios**: Although SSME performs well overall, the degree of error reduction is not as pronounced for some metrics (e.g., AUPRC) compared to others like ECE. Additionally, in subgroup analysis of AUC estimation, baseline methods such as Bayesian Calibration (BC) may outperform SSME under certain assumptions, such as when classifiers exhibit high AUC and monotonicity is assumed.

---

> ### Author Rebuttal · Authors · 2025-07-31
>
> Thank you for your comments! We appreciate that you believe SSME is able to utilize multiple sources of information simultaneously, has wide applicability, and outperforms baselines in estimating performance.
>
> We address your questions and respond to the highlighted weaknesses below.
>
> **Q1. Do the authors consider the selection of classifiers to be a practical issue that is adequately addressed in the paper, or do they believe it is straightforward enough not to pose a concern?**
>
> Thanks for this question. To address it, we ran an experiment where we measured SSME’s performance as the classifier set expands and the mix of classifiers within that set changes. In particular, for each number of classifiers between 2 and 7 (inclusive), we use SSME to estimate performance for each unique set of classifiers (so for 7 classifiers, there is exactly one set; for 4 classifiers, there are 7 choose 4 = 35 sets). We performed this experiment on the CivilComments dataset.
>
> As shown in the table below, we find that SSME’s performance improves as the number of classifiers increases, which accords with our theoretical results. As a caveat, our theory predicts that SSME is likely not to perform well if one adds an inaccurate (e.g., worse than random) classifier to the set.
>
> We also find that SSME's performance does not vary greatly across classifier sets, suggesting that SSME’s performance is robust to using different classifier sets. In particular, the standard deviation in accuracy estimation error across random classifier sets is always under 0.011, and the standard deviation in ECE estimation error is always under 0.015, indicating consistent performance across classifier sets.
>
> First, we find that performance does not vary greatly across classifier sets, suggesting that SSME’s performance is robust to using different classifiers. In particular, the standard deviation in accuracy estimation error across random classifier sets is always under 0.011, and the standard deviation in ECE estimation error is always under 0.015.
>
> Second, we find that SSME’s performance improves as the number of classifiers increases, which accords with our theoretical results. As a caveat, our theory predicts that SSME is likely not to perform well if one adds an inaccurate (e.g., worse than random) classifier to the set.
>
> | Number of classifiers |            MAE, ECE (SD) |       MAE, Accuracy (SD) |
> | --------------------: | -----------------------: | -----------------------: |
> |                     2 |          0.0457 (0.0059) |          0.0471 (0.0047) |
> |                     3 |          0.0302 (0.0107) |          0.0308 (0.0098) |
> |                     4 |          0.0268 (0.0094) |          0.0273 (0.0096) |
> |                     5 |          0.0252 (0.0072) |          0.0253 (0.0078) |
> |                     6 |          0.0235 (0.0043) |          0.0235 (0.0048) |
> |                     7 | 0.0225 (only one subset) | 0.0223 (only one subset) |
>
> (Dataset: CivilComments) MAE for ECE and accuracy across classifier set choice, along with the standard deviation in MAE across classifier sets.
>
> **Q2 + W2 On the assumption that labeled and unlabeled data are IID: do the authors believe that such distributional mismatch would not pose a serious issue in real-world applications? Alternatively, do they see this as a limitation that could be addressed through future methodological improvements?**
>
> Thanks for this question! We performed an experiment to assess SSME’s performance under violations of this assumption and found that SSME continues to outperform baselines at small amounts of labeled data. The experiment draws the unlabeled data for CivilComments from the unlabeled data provided by WILDS, which is specifically designed to exhibit realistic distribution shifts. We report the average error in accuracy estimation for each method. SSME achieves an average estimation error (in MAE, scaled to same 0-100 scale as the paper) of 2.12 (averaging across 20, 50, and 100 labeled datapoints), compared to 4.15 for labeled data alone and 4.44 and 4.56 for the two best-performing baselines (Bayesian-Calibration and Dawid-Skene). This demonstrates that SSME still provides an advantage over baselines even in the presence of real-world distribution shifts.
>
> Furthermore, we believe the assumption that the unlabeled data is drawn from the same distribution as the labeled data is realistic in several important settings. The assumption is met if the annotator/practitioner chooses samples to label at random. For instance, if the labeling budget is small, a practitioner may select a random subset to label. There are also settings where this assumption approximately holds. For instance, in radiology, if one considers the “labeled” images to be all images that have been labeled in the past (e.g., for a particular hospital) and the “unlabeled” images to be the images a radiologist has yet to look at, there is no reason to expect a distribution shift.
>
> **W1. Simplifying assumptions in theoretical analysis.**
>
> Thank you for this question. We use UL+ for the theoretical analysis as it enables us to build on prior results in semi-supervised learning for mixture models, which use this as a standard assumption. To our knowledge, there are no results establishing analogous performance bounds (i.e., bounds relating the amount of data to parameter estimation error) for EM algorithms. The bound derived from analyzing UL+ is likely conservative, as UL+ first uses the unlabeled data to identify decision boundaries then uses the labeled data to assign regions to classes. In general we expect EM, which directly maximizes the log-likelihood of both the labeled and unlabeled data, to perform even better.
>
> **W3. Performance variability in specific metrics and scenarios**
>
> The reviewer points out that, although SSME performs well overall, it reduces the error more for some metrics than for others. The reviewer is correct that SSME’s performance is strong, outperforming baselines across all metrics and tasks. In particular, SSME outperforms the next-best baseline, Bayesian-Calibration, in every metric tested (2.1x gain in accuracy estimation, 1.5x gain in ECE estimation, 1.2x gain in AUC estimation, and 1.1x gain in AUPRC estimation) averaged across tasks.
>
> We agree that the performance gains are not uniform across metrics. This behavior is consistent with our theory, which shows that error bounds on AUC are looser than those on accuracy (because the error bound on estimated AUC scales differently with separation, see the $\sqrt{2}$ denominator instead of 2 in Equations 3 and 4).
>
> Furthermore, the reviewer points out that under certain assumptions, Bayesian-Calibration might outperform SSME. We agree that Bayesian-Calibration may outperform all other methods if its strong assumptions on the classifier set (monotonic relationship between classifier scores and Pr(Y=1)) exactly hold. However, as our empirical results demonstrate, in real-world settings the assumptions Bayesian-Calibration requires do not hold well enough for it to outperform SSME, which achieves better performance for every metric.  Additionally, Bayesian-Calibration is restricted to settings with binary classifiers, a significant limitation that SSME addresses.
>
>
> **Please let us know if our response addresses your primary concerns; if so, we would kindly appreciate it if you could raise your score. If not, are there additional concerns we can address?**

---

> > ### Comment · Reviewer_Nk8N · 2025-08-07
> >
> > Thank you for the authors' very thorough rebuttal. Since you have addressed the reviewer's questions with additional data included, most of my concerns and questions have been resolved. Therefore, I am raising my evaluation by one level from 3 to 4.

---

> > > ### Author Response · Authors · 2025-08-07
> > > **Author reply**
> > >
> > > Thank you for engaging with our rebuttal and raising your score!

---

> ### Comment · Area_Chair_gjRT · 2025-08-06
>
> Dear Reviewer,
>
> The author have provided their rebuttal with additional results. Please kindly reply to the authors as soon as possible before the discussion period ends.
>
> Thanks a lot.
>
> Best regards,
>
> AC

---

### Official Review · Reviewer_Sq1D · 2025-07-03

**Clarity:** 3
**Significance:** 3
**Originality:** 3
**Rating:** 4
**Confidence:** 3

**Summary:**

This paper introduces a method called Semi-Supervised Model Evaluation (SSME) that uses both labeled and unlabeled data to evaluate machine learning classifiers. The key to SSME is to estimate the joint distribution of ground-truth labels and classifier scores using a semi-supervised mixture model. Besides, the authors provide theoretical results to show the upper bounds of estimation error. Experiments in four domains compared with eight baselines show the effectiveness of the proposed method.

**Questions:**

1. In Algorithm 1, do different initializations affect the estimations of parameters?
2. Why do you use a different algorithm UL+ in the theoretical analysis?
3. The error bounds are tightened as the number of classifiers increases, but adding classifiers also brings additional expenses. Is there an optimal number of classifiers when considering the trade-off between performance and computational expenses?
4. The authors provide results of the robustness of SSME to kernel choice in Table S13. How about the robustness of SSME to bandwidth h?

**Ethical Concerns:**

["NO or VERY MINOR ethics concerns only"]

**Final Justification:**

The issue of inconsistency between the theoretical analysis (UL+ algorithm) and the empirical application (EM algorithm) remains unsolved. I cannot strongly approve for this paper, and my recommendation is borderline accept.

**Limitations:**

Yes, the authors have adequately addressed the limitations.

**Paper Formatting Concerns:**

No formatting issue found in this paper.

**Quality:**

3

**Strengths And Weaknesses:**

Strengths:
1. This paper is well-written and easy to follow.
2. The proposed method is straightforward, and the optimization procedure is clear.
3. This paper presents theoretical results and in-depth analysis, which explains why SSME outperforms baselines.

Weaknesses:
1. The authors assume that the unlabeled samples are drawn from the same distribution as the labeled samples, which may violate reality.
2. The authors optimize the parameters using EM over 1000 epochs, which may be time-consuming. The running time or computational complexity should be provided.
3. Theorem 1 is derived by using the UL+ algorithm, however, the Algorithm 1 uses EM to estimate $P(y,s)$, which is not consistent.

---

> ### Author Rebuttal · Authors · 2025-07-31
>
> Thank you for your positive review and recognizing both our theoretical contributions and in-depth empirical analysis. We respond to your questions and suggestions below.
>
> **Q1. Do different initializations affect the estimation of parameters?**
>
> Good question. We find that SSME’s performance remains strong when using an alternative plausible initialization method: using a majority vote among k=5 nearest neighbors among the labeled examples to initialize cluster assignment for unlabeled examples. We compare this alternate initialization method to our original initialization method (which we refer to as "draw" in the table below) on the CivilComments dataset with three different amounts of labeled data (20, 50, and 100 points), just as in our original paper. We include our results on ECE and accuracy estimation error in the table below, where we find that the two initialization methods perform comparably on the CivilComments dataset (with KNN slightly beating the original initialization method in 2 of 3 settings, and original beating KNN in the other). Notably, we continue to outperform all the baselines even with the alternative KNN initialization. This is useful practical guidance for implementing SSME, which we will incorporate into the paper.
>
> | Number of labeled examples | Initialization method | MAE, ECE  | MAE, Accuracy |
> | -------------------------- | --------------------- | --------- | ------------- |
> | 20                         | KNN                   | 0.045     | 0.046         |
> | 20                         | draw             | 0.035 | 0.034     |
> | **—**                      | **—**                 | **—**     | **—**         |
> | 50                         |  KNN               | 0.037 | 0.040     |
> | 50                         | draw                  | 0.043     | 0.043         |
> | **—**                      | **—**                 | **—**     | **—**         |
> | 100                        | KNN              | 0.031 | 0.033     |
> | 100                        | draw                  | 0.037     | 0.036         |
> (Dataset: CivilComments) SSME’s performance is robust to choice of initialization. The original initialization method presented in the paper is denoted "draw."
>
>
> **Q2 + W3. Why use a different algorithm (UL+) in the theoretical analysis?**
>
> Thank you for this question. We use UL+ for the theoretical analysis as it enables us to build on prior results in semi-supervised learning for mixture models, which use this as a standard assumption. To our knowledge, there are no results establishing analogous performance bounds (i.e., bounds relating the amount of data to parameter estimation error) for EM algorithms. The bound derived from analyzing UL+ is likely conservative, as UL+ first uses the unlabeled data to identify decision boundaries then uses the labeled data to assign regions to classes. In general we expect EM, which directly maximizes the log-likelihood of both the labeled and unlabeled data, to perform even better.
>
> **Q3. Error bounds tighten as the number of classifiers increases; but adding classifiers introduces expenses. Given this tradeoff, is there an optimal number of classifiers?**
>
> Thanks for raising this question! Empirically, we find that the computational expense of an additional classifier is very small. Concretely, fitting SSME with nine classifiers takes 17.4 seconds; in comparison, fitting SSME with one classifier takes 14.3 seconds. We provide theoretical conditions under which an additional classifier provides benefit in Section C.1: specifically if the separation between components exceeds $\sqrt{d+1}$, where $d$ represents the dimensionality, any additional classifier whose accuracy is better than random will tighten the bound on performance estimation error.
>
> **Q4. Robustness of SSME to bandwidth h?**
>
> Thanks for this question. We ran an experiment to test the robustness of our results to the choice of bandwidth. In the paper, we use the Sheather-Jones algorithm [1] to automatically identify a bandwidth. We compare this approach to (1) another automated bandwidth selection algorithm (the Silverman algorithm [2]) and (2) two fixed bandwidths, chosen to be larger but within an order of magnitude of the Sheather-Jones selected bandwidth. We conduct experiments using the CivilComments dataset and 3 labeled dataset sizes (20, 50, and 100 labeled datapoints). While results remain strong in all cases, we achieve the best performance when using the Sheather-Jones bandwidth selection procedure, as is done in the original manuscript. We will incorporate these results into the manuscript.
>
> | Number of labeled examples | Bandwidth selection procedure | MAE, ECE  | MAE, Accuracy |
> | -------------------------- | ----------------------------- | --------- | ------------- |
> | 20                         | **Sheather-Jones**            | **0.024** | **0.023**     |
> | 20                         | Silverman                     | 0.040     | 0.038         |
> | 20                         | 1.0                           | 0.037     | 0.036         |
> | 20                         | 2.0                           | 0.046     | 0.044         |
> | **—**                      | **—**                         | **—**     | **—**         |
> | 50                         | **Sheather-Jones**            | **0.023** | **0.022**     |
> | 50                         | Silverman                     | 0.066     | 0.065         |
> | 50                         | 1.0                           | 0.062     | 0.062         |
> | 50                         | 2.0                           | 0.045     | 0.044         |
> | **—**                      | **—**                         | **—**     | **—**         |
> | 100                        | **Sheather-Jones**            | **0.022** | **0.021**     |
> | 100                        | Silverman                     | 0.049     | 0.047         |
> | 100                        | 1.0                           | 0.056     | 0.055         |
> | 100                        | 2.0                           | 0.034     | 0.032         |
> (Dataset: CivilComments) Sheather-Jones outperforms another automated selection method and two pre-determined bandwidths.
>
> [1] https://www.stat.cmu.edu/~rnugent/PCMI2016/papers/SurveyBandwidthSelectionWand.pdf
>
> [2] https://sites.stat.washington.edu/wxs/Stat593-s03/Literature/silverman-81a.pdf
>
> **W1: Assuming that unlabeled samples are drawn from the same distribution as the labeled samples may violate reality.**
>
> Thanks for this point, which we agree is important. We have added an experiment to explore how SSME performs under violations of this assumption. Specifically, we test SSME’s performance when the unlabeled data for CivilComments is drawn from the unlabeled split provided by the WILDS benchmark [1], which is specifically designed to reflect real-world distribution shifts. SSME achieves an average estimation error (in MAE, scaled to same 0-100 scale as the paper) of 2.12 (averaging across 20, 50, and 100 labeled datapoints), compared to 4.15 for labeled data alone and 4.44 and 4.56 for the two best-performing baselines (Bayesian-Calibration and Dawid-Skene). This demonstrates that SSME still provides an advantage over baselines even in the presence of real-world distribution shifts.
>
> Furthermore, we believe the assumption that the unlabeled data is drawn from the same distribution as the labeled data is realistic in several important settings. First, the assumption is met if the annotator/practitioner chooses samples to label at random. For instance, if the labeling budget is small, a practitioner may select a random subset to label. There are also settings where this assumption approximately holds. For instance, in radiology, if one considers the “labeled” images to be all images that have been labeled in the past (e.g., for a particular hospital) and the “unlabeled” images to be the images a radiologist has yet to look at, there is no reason to expect a distribution shift.
>
> [1] https://arxiv.org/abs/2112.05090
>
> **W2. Providing running time and computational complexity.**
>
> We will include a discussion of computational cost to accompany our empirical results; thank you for the suggestion! Specifically, we will include the following. Fitting SSME is computationally cheap and faster than the best baseline. For every dataset and experiment configuration reported in the paper, SSME can be fit in under 5 minutes, using only 1 CPU. SSME inherits the computational complexity of kernel density estimators (O(n^2)) and EM. In a matched comparison (using 20 labeled examples, 1000 unlabeled examples, and 7 classifiers) the next best-performing baseline (Bayesian-Calibration) takes on average 32.1 seconds to estimate performance, while SSME takes 21.5 seconds.
>
> **Please let us know if our response addresses your primary concerns; if so, we would kindly appreciate it if you could raise your score. If not, are there additional concerns we can address?**

---

> > ### Comment · Reviewer_Sq1D · 2025-08-05
> >
> > Thanks for the authors' detailed response, most of my concerns have been addressed. However,  the issue of inconsistency between the theoretical analysis (UL+ algorithm) and the empirical application (EM algorithm) remains unsolved. I understand that the authors use the UL+ algorithm for theoretical analysis convenience and use the EM algorithm for better performance, but the response is not convincing for me. I have decided to retain my score.

---

### Author Response · Authors · 2025-08-01
**Overall reply and summary**

We thank the reviewers for their helpful and constructive reviews! We appreciate their recognition of the problem we tackle as “vital yet relatively overlooked” and “important,” their view that SSME estimates performance “more precisely” than competing methods, and their positive comments on the paper’s clarity and theoretical soundness.


The reviewers also had a number of helpful questions and suggestions, all of which we have addressed both below and in individual replies. We believe these additional analyses fully address the reviewers’ concerns and provide additional evidence that SSME is a reliable, broadly applicable method for semi-supervised model evaluation. We summarize key responses below:


**Performance relative to Bayesian-Calibration**: Reviewers Nk8N and HJAd were concerned that Bayesian-Calibration might outperform SSME, either in particular settings or overall. To clarify, our results show that SSME decisively outperforms Bayesian-Calibration across the diverse datasets that we study. In particular, our results show that SSME outperforms Bayesian-Calibration in every metric tested (2.1x gain in accuracy estimation, 1.5x gain in ECE estimation, 1.2x gain in AUC estimation, and 1.1x gain in AUPRC estimation), averaging across datasets and tasks. (SSME also outperforms the eight other baselines we assessed.) Further, Bayesian-Calibration does not generalize to settings with more than two classes, a key limitation that SSME overcomes.


**Simplifying assumptions in theoretical analysis**: Reviewers Sq1D and Nk8N asked about simplifying assumptions made to arrive at our theoretical results. We use a simplified semi-supervised learning algorithm (UL+) and make a Gaussian assumption about the distribution of model predictions. Both of these assumptions are standard in prior work, and we extend them to our setting to tractably derive theoretical results. In addition, we verify through comprehensive experiments that these results also hold empirically even when our theoretical assumptions do not apply.


**Robustness to alternate method parameters and inputs**: Reviewer Sq1D asked how SSME’s performance responds to different initializations and different bandwidths, while reviewers Nk8N and HJAd asked about the importance of classifier set design, including whether one should be careful which classifiers to include and what the benefit of additional classifiers would be. We have included additional experiments to address these comments, finding that SSME maintains its performance even with other initializations. Furthermore, we show that our automated bandwidth selection procedure (Sheather-Jones) outperforms other plausible bandwidth selection procedures. Finally, consistent with our theoretical results, we show that performance improves as additional classifiers are used.


**Assuming that labeled & unlabeled data are sampled IID**: Reviewers Sq1D and Nk8N asked whether it is realistic to assume that the labeled and unlabeled data are drawn from the same distribution.  We provide examples showing why such settings commonly arise in important real-world scenarios. We also conduct an additional experiment to measure SSME’s performance when this assumption does not hold – i.e., under realistic distribution shifts between labeled and unlabeled data. We find that SSME retains its advantage over baselines even in the presence of distribution shifts.

---

### Note · Authors · 2025-08-12

We thank the reviewers and the area chair for the active and constructive discussion period. We are glad that at least three of four reviewers recommend acceptance (the fourth reviewer says they are “generally convinced”, but we are not sure what their updated score is). Reviewers described the paper as “well-written and easy to follow,” with a “new and interesting” approach that is widely applicable, and supported by “in-depth” analysis demonstrating that SSME outperforms the eight baselines we compare to.

We believe we have addressed all reviewer concerns and suggestions, specifically:

- Reviewers asked about the robustness of our results to various factors, including initialization, bandwidth choice, classifier set composition, data modality, and distribution shift between labeled and unlabeled data. We added experiments showing we continue to outperform baselines under all these variations.
- Reviewers asked whether SSME’s advantages over one baseline, Bayesian-Calibration, held across settings, and we clarified that SSME outperforms the eight baselines we assess – including Bayesian-Calibration – across all metrics.
- Reviewers asked about the simplifying assumptions in the theory, and we explained that they are standard in prior work and that results hold empirically even when assumptions are violated.
- Reviewer questions about the scalability of KDEs to high dimensions led us to test a normalizing flow parametrization in a higher-dimensional setting (ImagenetBG), which outperformed all baselines and the KDE parametrization. These results demonstrate that SSME can incorporate alternative estimators for P(y,s) to suit higher-dimensional data.

Taken together, we believe SSME is a state-of-the-art method for the important task of semi-supervised evaluation, further strengthened by addressing all reviewer feedback during the discussion period.

---

### Decision · Program_Chairs · 2025-09-17

**Decision:**

Accept (poster)

**Comment:**

This paper introduces SSME, a semi-supervised method for performance estimation without labeled data, based on modeling the joint distribution of predictions and labels using KDE and classifier mixtures.

Reviewers generally acknowledge the novelty and practical relevance of the problem setting and experimental validation. Several concerns were raised, including limited coverage of related work, the gap between theory and implementation, and scalability in high-dimensional settings. The authors provided a detailed rebuttal, and some reviewers acknowledged these responses and updated their scores accordingly. Some remained concerned about clarity, especially in the writing and appendix.

While certain aspects could benefit from further refinement, I find the paper addresses an important and under-explored problem with a well-motivated approach and strong empirical results. I recommend acceptance.